# Three rules govern thalamocortical connectivity of fast-spike inhibitory interneurons in the visual cortex

**Yulia Bereshpolova[1], Xiaojuan Hei[1], Jose-Manuel Alonso[1,2], Harvey A Swadlow[1,2]***

[1]Department of Psychological Sciences, University of Connecticut, Storrs, United States; [2]Department of Biological and Vision Sciences, State University of New York College of Optometry, New York, United States

**Abstract** Some cortical neurons receive highly selective thalamocortical (TC) input, but others do not. Here, we examine connectivity of single thalamic neurons (lateral geniculate nucleus, LGN) onto putative fast-spike inhibitory interneurons in layer 4 of rabbit visual cortex. We show that three 'rules' regulate this connectivity. These rules concern: (1) the precision of retinotopic alignment, (2) the amplitude of the postsynaptic local field potential elicited near the interneuron by spikes of the LGN neuron, and (3) the interneuron's response latency to strong, synchronous LGN input. We found that virtually all first-order fast-spike interneurons receive input from nearly all LGN axons that synapse nearby, regardless of their visual response properties. This was not the case for neighboring regular-spiking neurons. We conclude that profuse and highly promiscuous TC inputs to layer-4 fast-spike inhibitory interneurons generate response properties that are well-suited to mediate a fast, sensitive, and broadly tuned feed-forward inhibition of visual cortical excitatory neurons.

**\*For correspondence:**
harvey.swadlow@uconn.edu

**Competing interests:** The authors declare that no competing interests exist.

## Introduction

Neurons within the visual pathway form highly specific connections to preserve the precise retinotopic organization needed for visual acuity. Such connection specificity requires an active developmental process of synaptic pruning that limits the number of neurons receiving input from the same axon. In thalamus, each retinal afferent makes profuse retinogeniculate connections early during development but the connections are dramatically pruned at later developmental stages leaving only a small subset of neurons connected to the same retinal afferent (*Hamos et al., 1987*). Similarly, thalamic axons are dramatically pruned during cortical development leaving only a subset of neurons connected to the same thalamic axon in the mature visual cortex (*Alonso et al., 2001*; *Taylor et al., 2018*). However, the specificity of thalamocortical (TC) connectivity differs for different classes of neurons in layer 4 (L4) of sensory neocortex. For all cell types, a high degree of topographic precision is a necessary condition for TC connectivity, but in some systems other requirements are also stringently imposed. For example, synaptic connectivity between cells of the lateral geniculate nucleus (LGN) and simple cells of the feline visual cortex (V1) is highly dependent on precise retinotopic alignment, but also requires a similarity of a number of receptive field (RF) properties of the thalamic inputs and cortical target neurons (*Alonso et al., 2001*; *Hirsch and Martinez, 2006*; *Sedigh-Sarvestani et al., 2017*). This specificity results in a relatively low connection probability between retinotopically-aligned LGN neurons and L4 simple cells, and accounts, in part, for the orientation and direction selectivity of visual cortical simple cells (*Alonso et al., 2001*; *Lien and Scanziani, 2018*; *Sedigh-Sarvestani et al., 2017*). By contrast, the probability of a synaptic connection between a TC neuron and a topographically aligned cortical fast-spike inhibitory neuron is much higher. For example, a connection probability of ~2/3 is seen between ventrobasal thalamic neurons

and putative fast-spike interneurons (suspected inhibitory interneurons, SINs) in the aligned somato-sensory cortex of rabbits (*Swadlow and Gusev, 2002*) and rats (*Bruno and Simons, 2002*). In cat visual cortex, these inhibitory cells also respond to thalamic inputs with less selectivity than do the (presumptive) spiny simple cells (*Sedigh-Sarvestani et al., 2017*).

Here, we examined the synaptic connectivity between LGN concentric neurons (the most popu-lous cell-type in the LGN) and L4 SINs in rabbit V1 (*Zhuang et al., 2013*). We found that three neces-sary conditions, or 'rules', regulate the connectivity between these populations. One rule concerns the precision of TC retinotopic alignment, another concerns the peak amplitude of the postsynaptic local field potential (LFP) elicited near the interneuron by the specific LGN neuron under study, and a third concerns the latency of the specific SIN's response to visual stimulation and electrical stimula-tion of the thalamus. Together, these rules generate a highly accurate prediction (in 41 of 42 cases) of whether an LGN-SIN pair will be synaptically connected. Moreover, the dense, highly divergent/convergent TC network that results from the application of these 'rules' is consistent with the broadly tuned RF properties of many fast-spike inhibitory cortical neurons (*Bruno and Simons, 2002*; *Hirsch et al., 2003*; *Nowak et al., 2008*; *Swadlow and Gusev, 2002*; *Zhuang et al., 2013*). It is also consistent with the fast, potent, and broadly tuned feed-forward inhibition onto L4 spiny cells (*Bagnall et al., 2011*; *Cruikshank et al., 2007*; *Gabernet et al., 2005*; *Miller et al., 2001a*; *Miller et al., 2001b*; *Swadlow, 2003*; *Taylor et al., 2018*).

## Results

We studied the functional connectivity between 50 concentric LGN neurons and 47 L4 SINs (64 LGN-SIN pairs) that had RFs aligned, to various extents, with those of the LGN RFs. *Figure 1A* shows the recording situation. Once an LGN neuron was isolated and the RF was plotted, a recording elec-trode was placed in the retinotopically-aligned region of V1 (after appropriate mapping procedures). Cortical recording electrodes consisted of either a 16-channel laminar probe, aligned perpendicular to the cortical surface, or a single microelectrode, similarly aligned, that was slowly advanced into L4. A stimulating electrode was also located in the LGN for gross electrical stimulation of TC inputs. This was used to identify V1 SINs. As described in Methods, SINs were identified by the high-fre-quency spike discharge elicited by electrical stimulation of the thalamus (3 + spikes elicited at fre-quencies of >600 Hz). SINs also had spikes of very short duration. *Figure 1* (see also supplementary figure 1) illustrates how we measured the properties of L4 SINs, and the frequency distributions of these measures, compared with those of L4 simple cells and SINs, previously studied in awake rabbit V1 using the same methods (from *Zhuang et al., 2013*).

Once the LGN neurons and V1 SINs were identified, we used sparse noise stimulation (*Figure 1B*) to further assess their retinotopic alignment, spatiotemporal RFs and other RF properties. *Figure 1C* shows an example of the spatiotemporal RF maps of an LGN concentric cell and an L4 SIN, shown as a color maps in a series of time delays after stimulus onset. By definition, LGN concentric neurons yield only ON-center or OFF-center responses at short latencies. By contrast, the RFs of SINs con-sists of highly overlapping ON and OFF response zones, which usually occur at similar amplitudes and latencies (*Zhuang et al., 2013*). In this case, the LGN cell was ON-center (*Figure 1C*, top two rows) and the SIN responded at similar amplitude and time course for light and dark stimuli (*Figure 1C*, bottom two rows). *Figure 1D* shows the spatial RF maps for this LGN/SIN pair, taken at the temporal window that yielded the peak response for each cell. *Figure 1E* shows, for the same LGN neuron (above) and L4 SIN (below), how the ON and OFF responses (and their spatial distribu-tions) evolve over time. The LGN cell clearly responded earlier than the SIN, and the ON and OFF responses of the SIN occurred with roughly similar amplitude and time course.

To assess synaptic connectivity between LGN cells and L4 SINs, we employed cross-correlation analysis. In order to avoid stimulus-induced correlations, spontaneous spike trains were used in these analyses (e.g. *Bereshpolova et al., 2011*; *Bereshpolova et al., 2019*; *Swadlow and Gusev, 2001*; *Swadlow and Gusev, 2002*; *Zhuang et al., 2013*). *Figure 1F* shows, for this LGN/SIN pair, a strong, brief (~1 ms) increase in SIN spike probability beginning ~1.3 ms (with the peak at 1.7 ms) following the LGN spike. Such short-latency, brief increases in spike probability are a hallmark of TC synaptic connectivity as measured by extracellular cross-correlation of spike trains (e.g. *Reid and Alonso, 1995*; *Tanaka, 1983*; *Swadlow and Gusev, 2001*). The 'efficacy' of this connection (an index of the probability that an LGN spike will elicit a SIN spike) was 2.6%.

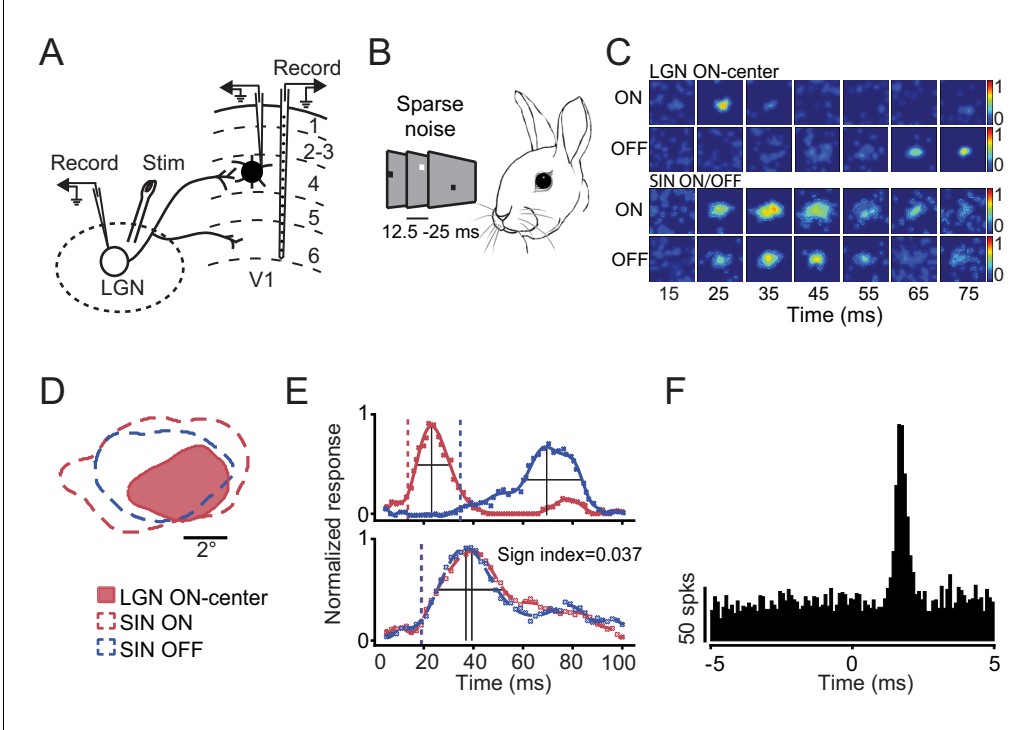

**Figure 1.** Experimental approach. (**A**) Schematic of the experimental preparation. Extracellular spikes were recorded simultaneously from one or more LGN cells and from one or more L4 SINs. Cortical recordings were obtained from either a 16-channel laminar probe (aligned perpendicular to the cortical surface) or from a single movable microelectrode. Electrical stimulation of the LGN was used to identify V1 SINs. (**B**) Mapping with the sparse noise stimulation allowed to obtain spatiotemporal RFs and ON or OFF subfields of LGN neurons and V1 SINs. (**C**) An example of the spatiotemporal RF maps of an LGN concentric cell and SIN of L4 shown as a color maps in a series of time delays after stimulus onset. (**D**) Spatial RF maps for the same LGN-SIN pair shown in C. The spatial maps were obtained by reverse-correlating neuronal responses with the white or black stimuli in a time window (±15 ms) around the response peak. Colors correspond to the response sign (red, ON; blue, OFF). (**E**) Temporal profiles of the spatial RF for the same ON-center LGN neuron (upper trace) and ON-OFF balanced SIN (lower traces) shown in C-D. Spatial response is normalized to maximum within each cell. Red and blue points are the average response at each time step and lines are a fitted polynomial function of ON and OFF responses respectively. The vertical dashed lines mark the latencies of RFs. Peak latency of the response is taken as the latency to the peak of the interpolating function (vertical lines), duration of the response is taken as the full width at half maximum value (horizontal lines). (**F**) Cross-correlograms for the same LGN-SIN pair indicating synaptic connectivity between LGN neuron and retinotopically-aligned SINs in L4. '0' on the x-axis represents the time of the LGN spikes. The correlograms are based on spontaneous activity.

The online version of this article includes the following figure supplement(s) for figure 1:

**Figure supplement 1.** Some characteristics of L4 SINs.

Retinotopic alignment was a dominant factor governing synaptic connectivity between LGN concentric cells and L4 SINs. *Figure 2A* shows, for all concentric LGN cells studied, how the likelihood of a synaptic connection with L4 SINs depends on retinotopic alignment. The proportion of connected pairs is high (31/42 cell pairs, ~73%) when alignment is within ½ of the diameter of the LGN RF center and drops off precipitously when misaligned by > ½ of a RF diameter. This implies a very high degree of divergence from single LGN neurons to multiple aligned SINs, and convergence from multiple LGN neurons to individual retinotopically-aligned L4 SINs. Importantly, however, although precise retinotopic alignment was a necessary correlate of synaptic connectivity, it was not sufficient. Thus, 11 of 42 (~26%) highly-aligned LGN/SIN pairs (where RF centers separated by <½ of an LGN RF diameter) showed no signs of synaptic connectivity. We initially thought that these retinotopically aligned but non-synaptically connected cases might be due to a dissimilarity in some LGN and cortical RF properties (as is the case for simple cells in cat visual cortex [*Alonso et al., 2001*]). For

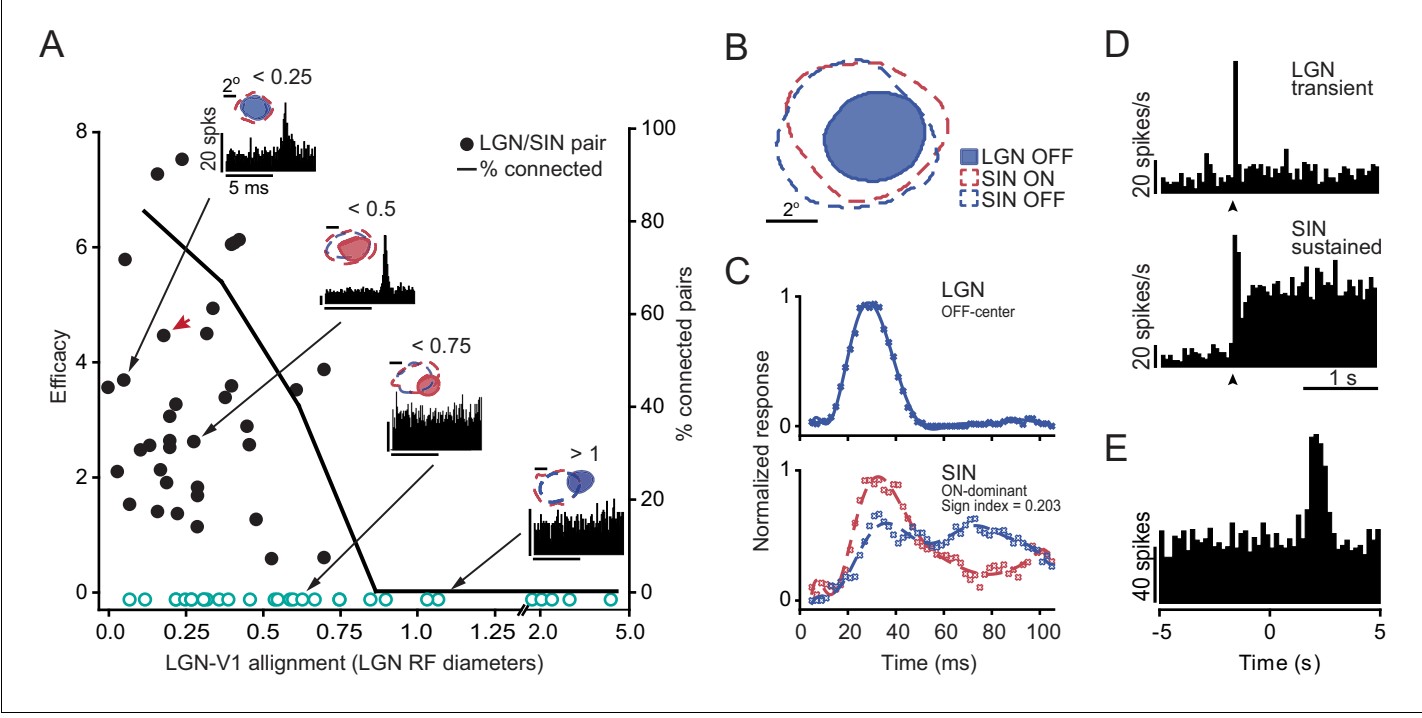

**Figure 2.** Rule 1: Retinotopic alignment. (**A**) Relationship between retinotopic alignment of LGN and L4 SIN RF centers (x-axis) and (1) the efficacy of the connection between concentric LGN neurons and SINs (filled circles, y-axis, left side) and (2) the percent connected cell pairs (solid line, y-axis right side). The proportion of connected pairs is high (~73%) when alignment is good (i.e. when the distance between thalamic and cortical RF centers is less than 1/2 the diameter of the LGN RF), and drops off precipitously as misalignment increases. Non-connected cell pairs were ascribed an efficacy of '0' and are shown by open circles. Insets show cross-correlograms for four LGN-SIN pairs with different degrees of RF alignment. Red arrow denotes the data point for cell pair shown in B-E. (**B-E**) Importantly, LGN-SIN synaptic connectivity does not depend on similarity of RF properties other than alignment. An example of connected cell pair with dissimilar RF sign (LGN cell was OFF-center, SIN was ON-dominated) and dissimilar sustained/transient cell class. (**B**) Spatial RF maps from a LGN neuron and a retinotopically-aligned L4 SIN. (**C**) LGN and SIN RFs are dissimilar in their RF sign. The LGN neuron yields only an OFF response at short latencies, and SIN yields earlier and stronger ON response. (**D**) These LGN and SIN RFs are also dissimilar in their sustained/transient responses. PSTHs of the responses of the LGN cell (upper) and the SIN (lower) to an optimal stationary stimulus placed over the RF center, presented for two seconds (arrows mark stimulus onset). The LGN cell responded in a purely transient manner, but the SIN's response had both transient and sustained components and was therefore classified as a 'sustained' cell (*Bezdudnaya et al., 2006*; *Stoelzel et al., 2008*; *Zhuang et al., 2013*). (**E**) Cross-correlation of the spike trains (spontaneous spikes only) of the LGN cell and the SIN, indicating synaptic connectivity.

example, the center responses of LGN concentric cells are either ON or OFF and, although most SINs show highly overlapping ON and OFF subfields, some are more dominated by ON or OFF responses.

*Figure 2B* shows such a case, where the RF maps from a concentric LGN neuron (OFF-center, yielding only an OFF response at short latencies), and from a very well-aligned L4 SIN (which yielded both ON and OFF responses at similar latencies). Post-stimulus time histograms (PSTHs) of the responses to light/dark stimuli are shown in *Figure 2C*, and show that the SIN response to the light stimulus was stronger than the response to the dark (OFF) stimulus. Thus, in this case, there was a clear dissimilarity between the RF properties of the LGN neuron (OFF-center) and SIN (ON-dominated). There was also a dissimilarity in the temporal dynamics of their response to a stationary stimulus (*Figure 2D*). LGN neurons can be classified as 'sustained' or 'transient' (*Cleland et al., 1971*; *Swadlow and Weyand, 1985*; *Bezdudnaya et al., 2006*; *Stoelzel et al., 2008*), and V1 SINs can also be classified in this manner (*Swadlow and Weyand, 1987*; *Zhuang et al., 2013*). In this case (*Figure 2D*), the LGN cell (upper histogram) responded in a transient manner, but the SIN's response (lower histogram) was sustained. Thus, this LGN/SIN pair, although very well-aligned retinotopically, was dissimilar with respect to both of these RF properties (ON/OFF and Sustained/Transient). Because of this, we thought that this cell pair would be one of those that were not

synaptically connected, but we were wrong. Cross-correlation of their spike trains revealed that this LGN-SIN cell pair was very well connected (*Figure 1E*), with LGN spikes generating a strong, brief increase in SIN spike probability beginning ~1.8 ms following the thalamic spike, and lasting about 1 ms. We found that neither similarity of LGN and SIN ON/OFF responses, nor similarity of sustained/transient responses was a significant factor in predicting the synaptic connectivity of retinotopically-aligned LGN/SIN pairs. Moreover, the probability of observing the connection in pairs where both parameters were similar did not significantly differ from pairs where both parameters were dissimilar ($X^2$ test, p=0.769).

## Two factors explain nearly all cases in which retinotopically-aligned, LGN/SIN cell pairs are not synaptically connected

In addition to retinotopic alignment, a critical factor controlling TC synaptic connectivity concerns the strength of the synaptic drive from the LGN cell to the local network within which the aligned SIN is imbedded. TC neurons can differ greatly in the overall synaptic drive that they produce at different depths within L4 (*Humphrey et al., 1985*; *Jin et al., 2008*; *Stoelzel et al., 2008*). Even neighboring LGN neurons may selectively target deep vs. superficial portions of L4 and some may not even project to V1 (e.g. interneurons). When recording in vivo, such differences are reflected by the amplitude of the spike-triggered LFPs and/or current source density (CSD) profiles generated by the spikes of single thalamic neurons at different cortical depths (*Jin et al., 2008*; *Jin et al., 2011*; *Stoelzel et al., 2008*; *Swadlow and Gusev, 2002*). To examine the effect of such differences, our cortical recording electrodes were filtered appropriately to enable recording both spikes (from the SINs and other cortical neurons) as well as low-amplitude LFPs that were triggered by the spikes of the retinotopically-aligned LGN neurons (which required methods of spike-triggered averaging to be revealed). We then determined how the amplitude of the monosynaptic component of the spike-triggered LFP generated near the SIN by spikes of the LGN neuron (*Swadlow and Gusev, 2001*; *Swadlow et al., 2002*; *Stoelzel et al., 2008*) was related to the probability of a synaptic connection of that LGN neuron with the SIN under study. *Figure 3A–C* shows an instructive case in which the RFs of two simultaneously recorded LGN neurons were both retinotopically aligned with the RF of a single SIN (*Figure 3A*) that was located superficially within L4. This SIN was recorded on the 8th channel of the laminar probe (shown by asterisk in *Figure 3B1 and B2*). One of these LGN neurons generated a maximum response more superficially in L4 (*Figure 3B1*, peak response at the same depth as the SIN), and this LGN neuron was synaptically connected to the SIN (*Figure 3C1*). The other LGN neuron generated a postsynaptic response that was deeper in L4 (*Figure 3B2*), with almost no response in superficial L4 (where the SIN was located). This LGN neuron showed no evidence of synaptic connectivity with the SIN (*Figure 3C2*), despite the good retinotopic alignment of their RFs. *Figure 3D* shows, for all 42 of our well-aligned LGN-SIN cell pairs, the significant relationship between the peak amplitude of the spike-triggered postsynaptic LFP response generated by the LGN neuron in the near vicinity of the aligned SIN, and the efficacy of the synaptic connection with the SIN. Non-connected cell pairs were ascribed an efficacy of '0' and are shown by open circles. Synaptic efficacies of '0' (no connection) were found for each of the 6 LGN neurons that failed to generate a postsynaptic LFP at the site of the SIN under study (represented by the large open circle, lower left). Thus, 6 of the 11 cases in which no synaptic connectivity was observed, despite excellent retinotopic alignment, can be accounted for by a very weak (or absent) LGN input to the vicinity of the SIN under study.

Next, we show that all-but-one of the remaining 'unexplained' cases of retinotopically aligned, but unconnected LGN-SIN pairs can be accounted for by a long-latency synaptic response of these 'unconnected' SINs to strong electrical stimulation of the LGN, and to visual stimulation. Thus, they appear to receive minimal direct input from LGN neurons. *Figure 4A* shows the distribution of synaptic latencies for SINs that were connected or were not connected to their aligned LGN neuron. None of the SINs responding at synaptic latencies of >3 ms to electrical stimulation of the thalamus were synaptically connected to their paired and well-aligned LGN neuron (31/37 of the SINs showing synaptic latencies < 3 ms were connected). Notably, SINs with long synaptic latencies to electrical stimulation of the LGN (>3 ms) also responded at longer latencies to visual stimulation than did those that responded at shorter synaptic latencies (<3 ms) (*Figure 4B*, 35.69 ± 1.78 ms vs. 25.77 ± 0.38 ms, respectively, p<0.001), suggesting that the long synaptic latencies of these SINs to electrical stimulation of the LGN reflect a fundamental difference in synaptic connectivity of these

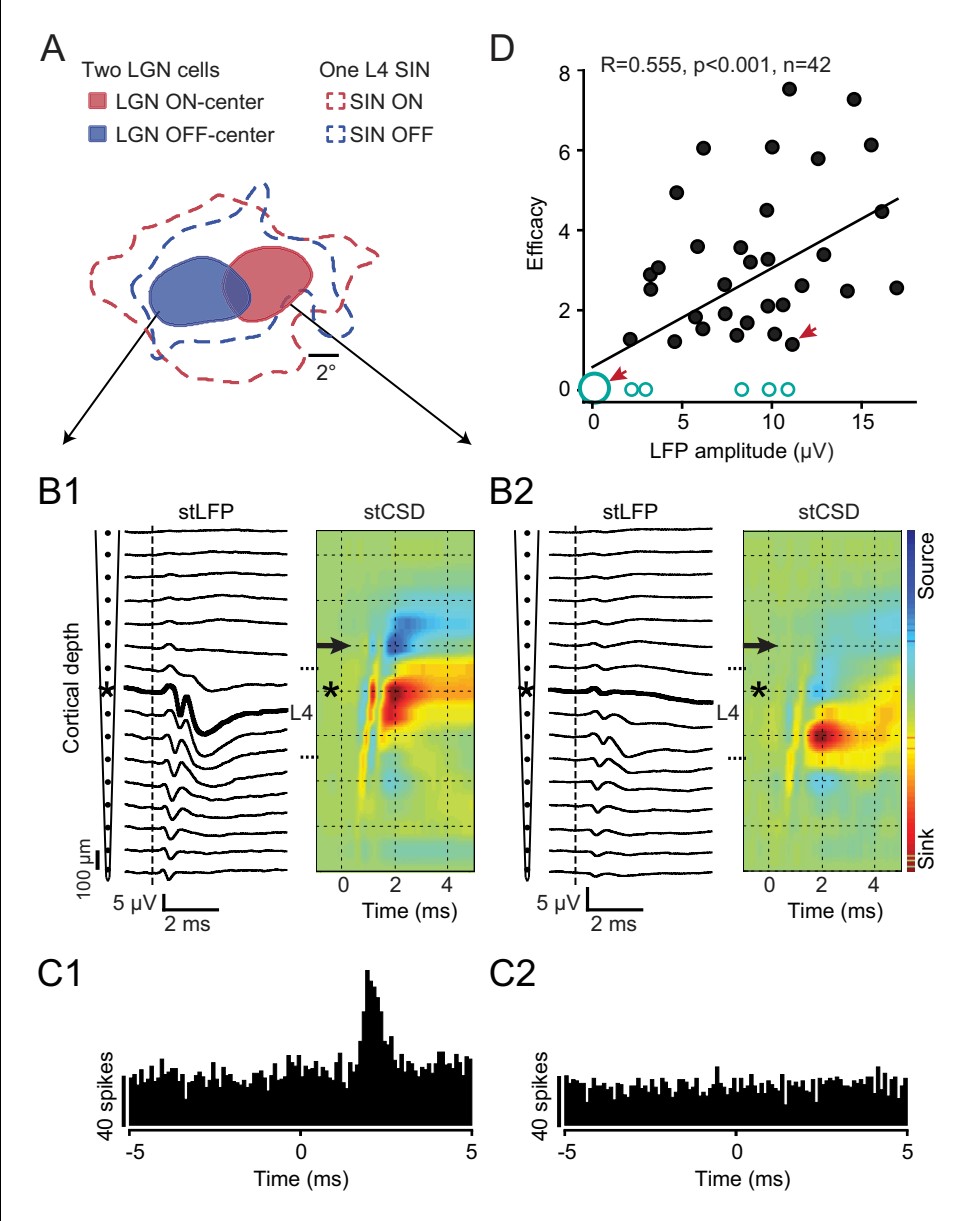

**Figure 3.** Rule 2: The 'strength' with which the LGN neuron provides a synaptic drive to the local network within which the aligned SIN is imbedded. (A-C) An example of the RF maps of two simultaneously recorded LGN neurons and a retinotopically-aligned SIN recorded on a linear probe. (A) The RFs of two LGN neurons (red - ON-center, blue - OFF-center) along with the RF of the SIN located within superficial L4 of V1. (B1, B2) Spike-triggered LFPs and the colorized CSD profiles generated in V1 by the spikes of these two LGN cells. The SIN was recorded on the 8th channel from the top of the probe (asterisk). The spike-triggered LFP/CSD generated by the OFF-center LGN cell had a maximum response more superficially in L4, at the same depth as the SIN, and this LGN neuron was synaptically connected to the SIN (C1). The ON-center LGN neuron generated a postsynaptic response that was deeper in L4 (B2), with almost no response in superficial L4 (where the SIN was located, and this LGN neuron was not connected to the SIN (C2)), despite the good retinotopic alignment of their RFs. (D) For all 42 well-aligned LGN-SIN cell pairs, the relationship between the amplitude of the spike-triggered postsynaptic response generated by the LGN neuron recorded near the SIN and the efficacy of the synaptic connection with the retinotopically-aligned SIN (non-connected cell pairs were ascribed an efficacy of '0' and are shown by open circles). The largest open circle illustrates LGN cells with no spike-triggered postsynaptic LFP and no connection with a SIN revealed by cross-correlation analysis. Red arrows denote data points derived from the LGN-SIN pairs shown in B1, C1 (right arrow) and B2, C2 (left arrow).

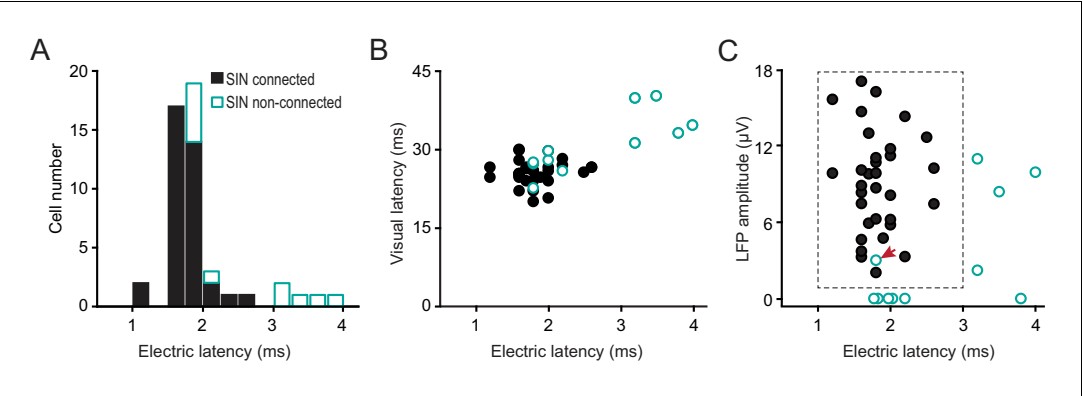

**Figure 4.** Rule 3: A third factor explains why some retinotopically-aligned LGN/SIN cell pairs were not synaptically connected despite strong retinotopic alignment and a strong postsynaptic LFP response generated by the LGN neuron in the vicinity of the SIN. This factor concerns the global synaptic connectivity of the specific L4 SIN with the LGN. Some L4 SINs appear to receive no (or very weak) direct synaptic input from the LGN, as indicated by a failure to respond at a short latency to either strong electrical stimulation of the LGN or visual stimulation. This accounts for the lack of connectivity of some well-aligned LGN-SIN pairs. (**A**) The distribution of synaptic latencies to electrical stimulation of the LGN, for SINs that were, or were not connected to their aligned LGN neuron. Note that none of the SINs responding at synaptic latencies of >3 ms were synaptically connected to its paired, well-aligned LGN neurons. (**B**) The relationship between synaptic latency to electrical stimulation of the LGN and the synaptic latency to visual stimulation. (**C**) For well-aligned connected and unconnected LGN-SIN pairs, the relationship between the amplitude of the postsynaptic LFP response generated by the LGN cell in the network surrounding SIN and the latency of the SIN to electrical stimulation of the LGN. These two variables account for all-but-one (indicated by the red arrow) of the cases in which the LGN/SIN pairs were not synaptically connected, despite precise retinotopic alignment.

cells with the visual thalamus. Importantly, the spike duration of these long-latency SINs did not differ from those of the other SINs (0.467 ± 0.022 ms vs. 0.449 ± 0.022 ms, p=0.337). Moreover, the long-latency SINs had similar RFs to those SINs that displayed short synaptic latencies (<3 ms) to electrical stimulation of the thalamus and were synaptically connected to one (or more) LGN cells. Thus, they all displayed spatially overlapping ON and OFF subfields, showed weak or no orientation selectivity, and responded in a non-linear manner to drifting grating stimulation (F1/F0 rations of <1). The depth within L4 of these SINs was also similar to the depths of the short-latency SINs. (199 ± 90 μm vs. 184 ± 22 μm, beneath the estimated superficial border of L4, p=0.822).

*Figure 4C* summarizes the extent to which (a) the amplitude of the postsynaptic LFP generated by the LGN neuron at the site of the SIN under study, and (b) latency of the SIN response to electrical stimulation of the thalamus account for all-but-one (small red arrow) of the cases in which the LGN-SIN cell pairs were not synaptically connected, despite precise retinotopic alignment. Thus, 31/ 32 well-aligned LGN-SIN cases were synaptically connected provided that (1) they received a measurable postsynaptic input from the LGN input (spike-triggered post-synaptic LFP amplitude >1 μV), and (2) the synaptic latency to electrical stimulation of the LGN was <3 ms. The only exception to these rules was a single LGN cell with a weak spike-triggered LFP amplitude that did not connect to a retinotopically-aligned SIN (1/31 well-aligned LGN-SIN pairs).

## Neighboring 'regular-spiking' neurons are more selectively connected to aligned LGN neurons than are SINs

Finally, the very high connection probability between retinotopically-aligned LGN/SINs pairs contrasts with the functional connectivity between LGN neurons and neighboring cortical neurons that were not SINs. We identified 28 neurons that were not SINs, but were located very near (in the same penetration and within 150 μm, either above or below) to a SIN that was synaptically connected to the same LGN neuron. Only three of these neurons (*Figure 5A–B*) received a functional input from the same LGN neuron (~11%) despite the fact that the spike-triggered LFP generated by the LGN neuron (recorded at the site of this non-SIN) was similar, albeit somewhat lower in amplitude than that measured at the site of the connected SIN (*Figure 5B*, 7.32 ± 0.59 μV and 8.72 ± 0.69 μV, respectively, p=0.023, paired t test). These non-SINs were 'regular-spiking', having considerably

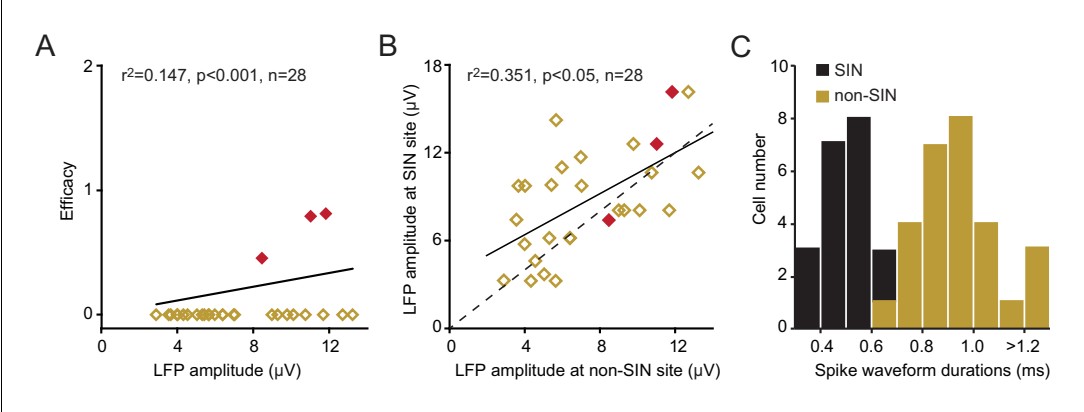

**Figure 5.** Functional connectivity between LGN neurons and L4 non-SINs. (A)The relationship between the amplitude of the spike-triggered postsynaptic response generated by the LGN neuron, recorded at the site of the non-SIN, and the efficacy of any synaptic connection with the non-SIN. Only three of these non-SINs (red) showed synaptic connectivity. Each of these non-SINs was located within 150 µm (either above or below) of a SIN that did receive a synaptic input from the same LGN neuron. (B) The relationship between the peak amplitude of the spike-triggered postsynaptic responses generated by a single LGN neuron, at the site of a synaptically connected SIN (y-axis) and at the site of a non-SIN (x-axis, located at a vertical distance of <150 µm from the connected SIN). For all data points, the SIN is connected to the LGN neuron, for red points, both the SIN and the non-sin are connected. (C) The frequency distribution of spike waveform durations for the L4 non-SINs shown in A and for the neighboring connected SINs. The L4 non-SINs had considerably longer-duration spikes than did the neighboring connected SINs.

longer-duration spikes (0.921 ± 0.028 ms) than the neighboring connected SINs (0.455 ± 0.022 ms, p<0.001, *Figure 5C*).

## Discussion

### The three rules governing TC connectivity onto L4 SINs

We show that three basic conditions must be met to ensure that an L4 SIN will receive a synaptic drive from an LGN concentric cell. The first, and most salient of these is retinotopic alignment. When this sole condition is met, 73% of LGN-SIN pairs display functional connectivity. The second necessary condition to ensure synaptic connectivity of LGN-SIN pairs concerns the strength of the postsynaptic LFP generated near the SIN by the specific LGN neuron under study. We looked only at the peak amplitude of the initial 1 ms of this postsynaptic response to ensure that we were examining a monosynaptic effect. The synaptic impact of some well-aligned LGN neurons may be stratified within L4 and be quite strong at some depths but weak at others (e.g. the case shown in *Figure 3*). It is not surprising, therefore, that an LGN neuron that generates little or no monosynaptic drive at the cortical depth of the L4 SIN would fail to make a synaptic connection with that SIN, even when retinotopic alignment is precise. In hindsight, this factor may appear obvious, for how could an LGN cell provide a synaptic drive to a SIN if it does not generate synapses near it? However, previous single-cell studies of TC connectivity (i.e. cross-correlation studies) have not taken a measure of the synaptic profile provided by the TC neuron to the region around the cortical cells under study. In previous cross-correlation studies, this factor has simply been unknown. Here, we gained a measure of this by examining the spike-triggered LFP generated near the SIN by spikes of the single LGN cell under study. This is a fairly easy measure to take in cross-correlation studies using extracellular microelectrodes, and we suggest that its general employment would reduce the variability seen in many cross-correlation studies of synaptic connectivity.

The third factor that must be met to ensure a connectivity between LGN neurons and retinotopically-aligned L4 SINs concerns the fact that some L4 SINs do not appear to receive a significant monosynaptic input from *any* LGN cells. We infer this from our finding that (1) these L4 SINs respond at very long synaptic latencies (>3 ms) to strong electrical stimulation of the LGN, (2) that these cells also respond at long latencies to visual stimulation, and (3) that cross-correlation analysis reveals no synaptically connectivity between these SINs and retinotopically-aligned LGN cells. Notably, these

cells responded, like other L4 SINs, with a burst of high-frequency spikes (at >600 Hz) to electrical stimulation of the LGN (albeit, at longer latencies), and their spike durations were as short as those of other SINs. They are, therefore, indistinguishable from other putative fast-spike interneurons, based on traditional extracellular measures (e.g. *Bruno and Simons, 2002*; *Zhuang et al., 2013*). Their estimated depth within L4 was also similar to other SINs. Together, these data support the notion that these 'long-latency' SINs represent a different sub-class of fast-spike inhibitory interneuron in L4 that is connected to the thalamus in a different manner than are those that respond at short synaptic latencies to LGN stimulation. Similarly, recent studies in several cortical systems indicate that fast-spiking parvalbumin-expressing interneurons may be divisible into multiple functional subclasses (*Dávid et al., 2007*; *Garcia-Junco-Clemente et al., 2019*; *Shin and Moore, 2019*). Our results are supportive of this notion. Surprisingly, these 'long-latency' SINs, which seem to lack significant monosynaptic input from the LGN have RF that are indistinguishable from those of SINs which do receive strong monosynaptic LGN input. This, of course, raises the question of how these RFs are synthesized.

## The synthesis of RFs in L4 SINs

L4 SINs in rabbit V1 can be divided into two classes: those that respond at short latencies to strong electrical stimulation of the LGN (<2 ms, short-latency SINs), and those that respond at long latencies (>3 ms, long-latency SINs, *Figure 4*). Aside from the latency of the visual response, the RFs of these two cell classes are virtually identical, consisting of overlapping ON and OFF zones, and poor or no orientation/direction selectivity, and non-linear responses to drifting visual gratings. By contrast, most regular-spiking neurons (non-SINs) in L4 have 'simple' RFs consisting of one or more spatially separate ON and/or OFF zones, marked orientation/direction selectivity, and linear responses to drifting visual gratings (*Zhuang et al., 2013*). Our results show that, virtually all of the short-latency (first-order) L4 SINs receives a highly convergent input from virtually all of the LGN axons that synapse in their vicinity (*Figure 4*). Thus, the short-latency SINs receive monosynaptic input from ON- and OFF-center LGN axons, as well as from LGN cells with both sustained and transient response properties. The overlapping ON and OFF responding, and the broadly tuned response properties of these SINs are what might be expected to result from such a promiscuous TC input.

A more puzzling question concerns how RFs are synthesized in the 'long-latency' L4 SINs. Of course, these SINs might receive some monosynaptic thalamic input (either subthreshold, or very long-latency) despite the fact that strong and synchronous electrical stimulation of the LGN failed to generate short-latency spikes in these cells. Notably, there is little evidence for LGN axons with long conduction times in rabbit, as the vast majority of LGN neurons have TC conduction times of <2 ms (~98%, *Stoelzel et al., 2008*; *Swadlow and Weyand, 1985*). Of course, one possibility is that these SINs receive their input from a yet-to-be-discovered slowly conducting LGN pathway. Alternatively, RFs of the long-latency SINs may be synthesized from a highly convergent input from neighboring spiny (simple) cells. Thus, in L4 of somatosensory barrel cortex, fast-spike interneurons receive a powerful and highly convergent input from neighboring spiny (regular-spiking) neurons. This input (spiny to fast-spike) is twice as strong and seven times more probable than is the input to other neighboring spiny neurons (*Beierlein et al., 2003*). In V1, the L4 spiny neurons are largely simple cells. So, we could speculate that the 'long-latency' SINs synthesize their ON/OFF and broadly tuned RFs from such strong and highly convergent input from neighboring simple cells. The result would be a SIN RF with spatially overlapping ON/OFF zones (because the simple cells have both ON and OFF subfields) and minimal orientation tuning (because all orientations are represented by the neighboring simple cells and there are no orientation columns in Rabbit V1). Such disynaptic excitatory input would also be expected to shape the RFs of the short-latency SINs, which would have RFs that are synthesized both from the highly convergent monosynaptic LGN inputs *and* from the (disynaptic) input from neighboring spiny neurons (*Beierlein et al., 2003*), resulting in the same RF structure and lack of orientation tuning, but a shorter latency to visual stimulation and electrical stimulation of the thalamus.

## The sparseness of TC connections to L4 regular-spiking (presumptive spiny neurons)

By contrast to the profuse TC connectivity seen in L4 SINs, regular-spiking neurons (presumptive spiny neurons), found just above or below an L4 SIN that is connected to an LGN neuron (within 150 microns in the same vertical penetration) have a much lower connection probability (~11%, compare *Figures 5A* and *3D*), despite the proven efficacy of the tested LGN neuron in driving the neighboring SIN. We have previously shown that > 80% of such regular-spiking non-SINs in L4 of rabbit V1 are simple cells (*Zhuang et al., 2013*). Similarly, in L4 of cat V1, simple cells receive a highly selective input from LGN TC neurons that depends on retinotopic alignment and spatiotemporal RF match, while the inputs to fast-spiking neurons are less selective (*Alonso et al., 2001*; *Reid and Alonso, 1995*; *Sedigh-Sarvestani et al., 2017*). Findings in somatosensory barrel cortex of rats are similar, where regular-spiking L4 neurons receive input from ventrobasal TC neurons at a much lower probability than do L4 fast-spike neurons (*Bruno and Simons, 2002*).

## Comparing the RFs of rabbit L4 SINs with those of other species

These results, as well as previous results in rabbit V1, indicate that the RFs of putative fast-spike GABAergic inhibitory interneurons form a uniform class of 'complex' cells, with spatially overlapping ON and OFF subfields and little or no orientation or directional selectivity. In the cat, however, the RF properties of fast-spike interneurons are more heterogeneous. Thus, *Hirsch et al., 2003* recorded intracellularly from 10 neurons with smooth dendrites and presumptive fast-spiking properties. They found that four of these cells had RFs that closely resembled the SINs of this study (spatially overlapping ON/OFF subfields, little orientation/direction selectivity), and the remaining 6 of these cells had classic 'simple' RFs. Similarly, *Nowak et al., 2008* recorded intracellularly and used cluster analysis to segregate cell types based on electrophysiological and RF properties. They found that some fast-spike cells were very broadly tuned to orientation, and they concluded that this class of fast-spike cell was "... likely to correspond to the non-orientation selective, complex inhibitory neurons of layer four described by *Hirsch et al., 2003*". By contrast, *Cardin et al., 2007* found only small differences between fast-spike and regular-spike cell classes in cat L4. Thus, it seems that the SINs found in rabbit V1 (*Swadlow and Weyand, 1987*; *Swadlow, 1988*; *Zhuang et al., 2013*; this study) may have their counterparts in feline V1, but that the well-tuned fast-spike interneurons with 'simple' RFs found in cat L4 are not found in rabbit V1. These cells have not yet been studied in L4 of mouse (or monkey) V1. In layer 2/3 of mouse, however, inhibitory interneurons of multiple classes were found to be very broadly tuned to orientation and a number of other stimulus features (*Kerlin et al., 2010*; *Sohya et al., 2007*). We would speculate that in the mouse (and other rodents), the RF properties of L4 fast-spike neurons will prove to be very similar to those found in rabbit V1.

## Conclusions

The promiscuous connectivity seen here between LGN concentric neurons and retinotopically-aligned L4 SINs is reminiscent of the 'complete transmission line' between nodes of a network, described by *Griffith, 1963*, and elaborated by *Abeles, 1991* (who adapted such a network as the basis of his 'synfire chain'). In such networks, each member of one node excites each member of the successive node via richly convergent/divergent synaptic connectivity. Neurons in the output node (SINs, in this case) are expected to show high sensitivity (because of their profuse inputs), but they suffer a decreased ability to discriminate among any selective properties of the input elements. Therefore, neurons in the output node will be broadly tuned to stimulus features represented by cells at the input node. These network characteristics nicely describe many of the response properties of L4 SINs in rabbit V1 (*Zhuang et al., 2013*). Thus, these cells are more sensitive to stimulus contrast than are L4 simple cells and have a spatial RF that consists of spatially overlapping ON and OFF subfields (despite the fact that the concentric LGN input cells responded to either ON or OFF in the RF center). Moreover, they are more broadly tuned to a number of stimulus parameters (spatial and temporal frequency, stimulus direction and orientation) than are simple cells of V1.

Similarly, SINs of somatosensory barrel cortex are synaptically connected to neurons in ventrobasal thalamus in a richly divergent/convergent manner. In that system, cross-correlation studies reveal a connection probability of ~2/3 between SINs in an L4 barrel, and TC neurons of the topographically aligned thalamic barreloid (*Bruno and Simons, 2002*; *Swadlow et al., 2002*). Notably,

SINs of S1 have RF properties that are highly analogous to those of V1, and their RF response properties are highly consistent with the expected properties of the 'complete transmission line' of *Griffith, 1963*. Thus, they display 'high sensitivity', showing lowest thresholds to whisker stimulation of any neurons found in the S1 barrel, and they show very broad temporal and directional tuning (*Swadlow, 1989*; *Swadlow, 1995*). The lack of directional tuning in SINs of L4 barrel cortex is thought to result from the highly convergent input to these from multiple neurons in their aligned thalamic barreloid that display different directional preferences (*Swadlow and Gusev, 2002*).

These results have implications for our current understanding of TC development and function. They suggest that TC pruning during development is likely to be more pronounced in excitatory than in fast-spike inhibitory cortical neurons leading to differences in their RF selectivity (*Alonso and Swadlow, 2005*; *Cardin et al., 2007*; *Bruno and Simons, 2002*). This connectivity difference is likely to generalize across sensory systems and species. For example, in both visual and somatosensory cortex, each thalamic afferent is known to make connection with a small percentage of cortical excitatory neurons that are retinotopically or somatotopically aligned, a percentage that is estimated to be ~15 to 33% in cat visual cortex (*Alonso et al., 2001*; *Sedigh-Sarvestani et al., 2017*), 11% in rabbit visual cortex (as shown here, *Figure 5A*), and 37% in rat barrel cortex (*Bruno and Simons, 2002*). Also, in both visual and somatosensory cortex, each thalamic afferent makes a higher percentage of connections with fast-spike inhibitory than excitatory neurons (*Swadlow and Gusev, 2002*, *Alonso and Swadlow, 2005*; *Bruno and Simons, 2002*; *Sedigh-Sarvestani et al., 2017*). While these similarities in TC connectivity are well known, all estimates of connection probability reported in the past failed to take into account the laminar specificity of the axonal arbor from the thalamic afferent and the presence of L4 cortical neurons that may receive weak or no direct thalamic input.

Remarkably, our results indicate that, when the laminar specificity of the axonal arbor is taken into account, each thalamic afferent makes connection with nearly all of the first-order (short-latency) fast-spike inhibitory neurons that are within reach of the axonal arbor regardless of the response properties of the LGN and cortical neurons. This is clearly not the case with neighboring L4 regular-spiking neurons. The 'three rules' of synaptic connectivity that we describe are both necessary and sufficient for predicting connectivity between LGN neurons and L4 SINs (40/41 cases, *Figure 4C*). Together, they predict connection probability with a surprising high level of precision that is not yet possible for any other TC circuit in any other species. This nonselective pooling of thalamic afferents can be amplified by nonselective pooling of inhibitory neurons connecting to the same cortical excitatory neuron (*Taylor et al., 2018*).

In conclusion, we have shown that concentric LGN TC neurons form functional synaptic connections with retinotopically-aligned L4 SINs at a very high probability (~73%). This probability increases to nearly 100% when considering two additional factors: whether the aligned LGN TC neuron synapses at the depth in L4 where the SIN is located, and whether the aligned L4 SIN receives short-latency input from any LGN cells. Thus, virtually every first-order fast-spike interneuron in layer four receives input from nearly all of the LGN axons that synapse nearby (40/41 cases). We propose that the observed promiscuous, highly convergent TC connectivity onto retinotopically/topographically aligned L4 fast-spike inhibitory interneurons may be a general feature of sensory neocortex, and that this feature plays a causal role in generating the fast, sensitive, broadly tuned, and powerful feed-forward TC inhibition of excitatory neurons that is seen within L4 (*Bagnall et al., 2011*; *Cruikshank et al., 2007*; *Cruikshank et al., 2010*; *Gabernet et al., 2005*; *Hull et al., 2009*; *Miller et al., 2001b*; *Swadlow, 1995*; *Swadlow, 2003*; *Taylor et al., 2018*).

## Materials and methods

### Experimental procedures and data analysis

Extracellular single-unit recordings were obtained from LGN neurons and from the retinotopically-aligned region of V1 in four awake adult female Dutch-Belted rabbits. The general surgical procedures for chronic recordings and anesthesia have been described in detail previously (*Swadlow et al., 2002*; *Bereshpolova et al., 2007*; *Stoelzel et al., 2008*; *Zhuang et al., 2013*) and are reported only briefly here. All animal procedures were conducted with the approval of the University of Connecticut Animal Care and Use Committee in accordance with the National Institutes of Health guidelines.

Initial surgery was performed under ketamine-acepromazine anesthesia using aseptic procedures. After removal of the skin and fascia above the skull, the bones of the dorsal surface of the skull were fused together using stainless steel screws and acrylic cement. A stainless steel rod (6 mm in diameter, thinned to 2 mm in places to conserve space on the skull) was oriented in a rostrocaudal direction and cemented to the acrylic mass. The rabbit was held rigidly by this rod during later surgery and recording sessions. The silicone rubber was used to buffer the wound margins and was covered with the acrylic cement to create base for an attachment of chronically implanted equipment. A layer of acrylic cement also always covered the exposed skull between recording sessions.

A concentric array of seven independently movable electrodes was placed within the LGN (electrode separation ~200 um, *Swadlow et al., 2005*). Recordings were made using fine-diameter (40 micron) quartz-insulated platinum/tungsten electrodes tapered and sharpened to a fine tip. Extracellular single-unit recordings were recorded from the retinotopically-aligned region of V1 using either 16-channel silicon probes with 100 microns vertical spacing (NeuroNexus Technologies) or the same fine-diameter single electrodes, that were moved through the depth of the cortex. All electrophysiological activity was recorded in the awake state (e.g. *Bereshpolova et al., 2007*; *Bereshpolova et al., 2019*; *Bezdudnaya et al., 2006*; *Zhuang et al., 2013*) and acquired using a Plexon data acquisition system (Plexon, Dallas, TX). Signals were amplified, bandpass filtered and sorted to identify single units. LFPs were also recorded, filtered at 2 Hz to 1.9 kHz (half-amplitude), and sampled continuously at 10 kHz.

## Visual stimulation and RFs analysis

All stimuli were generated using custom-made program (Visual C++, DirectX 7), and presented on a CRT monitor (Nec MultiSync 40 × 30 cm, mean luminance 48 cd/m2, refresh rate 160 Hz). RFs were mapped using sparse noise (*Jones and Palmer, 1987*), made of white and black squares (0.5–2° in a grid of 30 × 22 degrees) in a pseudorandom sequence, and matrices of the raw ON and OFF RF data were generated by reverse correlation method. A Gaussian filter and bicubic interpolation were applied to facilitate visualization. The luminance of gray background was adjusted so the contrasts from the white and black squares were equal.

Spatial RF maps were calculated for a series of time delays (using sliding time window with 1 ms step and 10 ms duration) between stimulus and response. For each time delay after stimulus onset, the spatial RF was computed by averaging spikes that follow the presentation of the stimulus at each position of the grid. The average spatial responses at each time delay were fit with a polynomial function. From these fits, we extracted temporal RF parameters such as latency of visual response (defined as the time at which the variance first crosses the three times SD above the mean variance of noise at delay 0–10 ms after stimulus onset), peak latency of the response (measured as the time at which the interpolating function reached maximum value), duration of the response (defined as the full width of the interpolating function at half maximum value).

To assess whether the spatial RF in the restrict peak area of SINs was dominated by ON or by OFF subfields, a 'sign index' was calculated (*Van Hooser et al., 2013*):

$$\mathrm{Sign\,index} = \frac{\left| \sum R_{ON(i,j)} - \sum R_{OFF(i,j)} \right|}{\sum R_{ON(i,j)} + \sum R_{OFF(i,j)}},$$

Where $R_{ON(i,j)}$ is the response to all bright squares at position i,j and $R_{OFF(i,j)}$ is the response to all dark squares at position i,j. The sign index ranged from 0 to 1, with 0 representing a balanced spatial RF, and one an ON-dominated or OFF-dominated spatial RF.

The sign index and response latency of RF were used to define the strongest subregion of cortical cell for assessing the degree of spatial overlap with LGN RF.

## Assessing retinotopic alignment

To achieve precise retinotopic alignment between thalamic and cortical recording sites, as a first approximation, cortical RF maps were obtained from the multiunit activity recorded at multiple depths within the cortex using a single movable microelectrode. Based on the known topography of V1, the axis of the mapping electrode was adjusted by retracting and reinserting it at a slightly different angle until the recording sites at different depths were very well-aligned with each other, and with the LGN neuron, based on their highly overlapping RF locations. In some cases, the mapping

electrode was replaced with the 16-channel probe. Once RFs were tested and mapped, a large number of spontaneous spikes (usually several thousand from each neuron) was recorded.

To quantitatively evaluate a spatial relationship between thalamic and cortical RFs, spatial RF structures at the peak response latency were reconstructed by calculating the center of mass of the response in visual space using the absolute value of all significant pixels in each RF. Then RF maps of both sites were fitted with an ellipse. Ellipse parameters such as width, height and elongation angle were extracted using the covariance matrix, eigenvalues and eigenvectors of a complex Hermitian. This approach provided the center position and aspect ratio of the spatial RFs. The distance between the spatial location of the RF center of the LGN cell and the center of the SIN RF was calculated based on ellipse parameters (width and height defined by 30% of the peak value). This distance was normalized to the RF diameter of the thalamic neuron.

## Cells and layers identification

Identification of cortical layers was based on the reversal point of the field potential generated by a diffuse flash stimulus (*Stoelzel et al., 2008*). SINs were identified by firing a burst of 3 or more high-frequency spikes (>600 Hz) to electrical stimulation of the LGN (*Swadlow, 1989*; *Swadlow, 2003*; *Zhuang et al., 2013*). SINs also had spikes of very short duration (*Figure 1—figure supplement 1*).

Sustained/transient identification: A flashing stimulus was presented over each cell's RF (at least 1 s on, 1 s off). Baseline activity was calculated as the average firing rate of the neuron for the period preceding the start of the stimulus for a half a second. Maintained discharge was calculated from the average firing rate from 0.5 to 1 s after stimulus onset. Concentric cells of LGN, which had a maintained response above baseline of more than 10 spikes per second, were classified as sustained (*Bezdudnaya et al., 2006*; *Cano et al., 2006*). The cortical sustained SINs were required to have a maintained response at least two times higher than baseline. Cells that had a ratio lower than two were classified as transient (*Zhuang et al., 2013*). Note that whereas 'transient' neurons have only a transient response component, 'sustained' neurons may have both.

## LFP and CSD analysis

Methods and rationale for localizing the pre-and postsynaptic responses generated by single TC neurons through the depth of the cortex have been described previously (*Swadlow et al., 2002*; *Stoelzel et al., 2008*; *Jin et al., 2011*). Spontaneous extracellular single-unit activity of the LGN cells was recorded along with the LFPs generated through the depths of the retinotopically-aligned region of V1. Spike-triggered averages of LFPs were generated from spontaneous LGN spikes. One-dimensional CSD profiles were computed from the voltage traces of the field profiles according to the method described by *Freeman and Nicholson, 1975*. Estimates for the CSD at the top and bottom recording sites were provided by the method of *Vaknin et al., 1988*. In the CSD profiles and their color maps, current sinks (red) are indicated by downward deflections and sources (blue) are indicated by upward deflections.

In order to assess the extent of the postsynaptic response generated by the LGN cell near the SIN under study ('Rule 2' in *Figure 3*), we measured the spike-triggered average responses generated on the probe channel (or single electrode) on which the SIN was recorded. An LGN neuron was considered to have generated a postsynaptic impact at the depth of recorded SIN after fulfilling the following two conditions (1) presence of a clear axon terminal potential ≥0.75 μV in amplitude; and (2) the postsynaptic response had to follow this axonal component by <1 ms and consist of a sharp negativity in the spike-triggered LFP of >1 μV in amplitude. To be sure that we were measuring the amplitude of the monosynaptic spike-triggered LFP, we measured the peak amplitude of this response during the initial 1 ms following the onset of the postsynaptic response.

## Assessing TC connectivity

The connectivity within simultaneously recorded geniculocortical pairs of neurons was assessed by using the cross-correlation analysis. Monosynaptic connection was inferred from the presence of a significant peak in the probability of SIN firing at intervals of 1–4 ms (reflecting the presynaptic axonal conduction time, the synaptic delay, and the rise-time of postsynaptic firing). A peak in a cross-correlogram was defined as significant when at least two of three successive bins (0.1 ms bin width) in the peak exceeded the 0.01 confidence level.

After detecting a significant peak in the cross-correlogram, we determined an efficacy value (*Levick et al., 1972*) based on a brief window (+0.6 ms) centered on the peak. Efficacy was calculated by counting the number of SIN spikes occurring during this window, subtracting the baseline number of expected spikes, and dividing this value by the number of triggering TC spikes. The number of SIN spikes expected by chance within each bin was based on the mean number of spikes per bin that occurred during a 5 ms window, from 4 ms before to 1 ms after the TC spike.

All the p values provided in Results represent the results of independent sample t test, if not specified. Data are provided as mean ± SEM.

## Additional information

### Funding

| Funder | Grant reference number | Author |
| --- | --- | --- |
| National Institutes of Health | R01EY028905 | Harvey Swadlow |

The funders had no role in study design, data collection and interpretation, or the decision to submit the work for publication.

### Author contributions

Yulia Bereshpolova, Conceptualization, Data curation, Software, Formal analysis, Supervision, Validation, Investigation, Visualization, Methodology, Writing - original draft, Project administration, Writing - review and editing; Xiaojuan Hei, Software, Formal analysis, Validation, Investigation; Jose-Manuel Alonso, Conceptualization, Methodology, Writing - review and editing; Harvey A Swadlow, Conceptualization, Resources, Data curation, Supervision, Funding acquisition, Validation, Visualization, Methodology, Writing - original draft, Project administration, Writing - review and editing

### Author ORCIDs

Yulia Bereshpolova (iD) http://orcid.org/0000-0002-7117-7255
Harvey A Swadlow (iD) https://orcid.org/0000-0003-1477-3250

### Ethics

Animal experimentation: All animal procedures were conducted with the approval of the University of Connecticut Animal Care and Use Committee i(Protocol No. A19-040) n accordance with the National Institutes of Health guidelines.

### Decision letter and Author response

Decision letter https://doi.org/10.7554/eLife.60102.sa1
Author response https://doi.org/10.7554/eLife.60102.sa2

## Additional files

### Supplementary files

• Transparent reporting form

### Data availability

All data generated or analysed during this study are included in the manuscript and supporting files.

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
