## [Decision Letter]

Thank you for submitting your article "Three rules govern thalamocortical connectivity of fast-spike interneurons in the visual cortex" for consideration by *eLife*. Your article has been reviewed by three peer reviewers, one of whom is a Guest Reviewing Editor, and the evaluation has been overseen by John Huguenard as the Senior Editor. The following individual involved in review of your submission has agreed to reveal their identity: Barry Connors (Reviewer #2).

The reviewers have discussed the reviews with one another and the Reviewing Editor has drafted this decision to help you prepare a revised submission.

Summary:

This manuscript shows, in the visual system of the adult rabbit, that LGN neurons and V1 suspected interneurons (SINs) in L4 show positive cross-correlograms when (1) RFs overlap spatially more than 50%, (2) LGN spikes trigger a measurable LFP response near the recorded SIN and (3) SIN have short latencies to electrical (< 3ms) and visual (< 30 ms) stimulation. SINs with longer latencies (n=5) supposedly do not receive LGN input and do not show positive cross-correlograms even with 100% spatial overlap. Thus, the results show that direct synaptic connections between LGN cells and SINs are predicted by three properties of the cells involved, and are insensitive to other response features. The study represents another important step forward in the understanding of thalamocortical transmission of sensory signals to neocortical layer 4.

Revisions:

1) The three reviewers agreed that the article is clearly written, well-documented and illustrated and very pleasant to read, the methodology is powerful and flawless, the results are convincing. The study represents another important step forward in the understanding of thalamocortical transmission of sensory signals to neocortical layer 4. However, the three reviewers agree on the confusing nature of the terminology pre and postsynaptic and the need for additional analysis of non-SIN neurons.

2) Issue regarding terminology:

One reviewer is concerned that the term mismatch applied to the sign of the RF and the time course of the response seems misleading. For example, I don't see a mismatch in time course in Figure 2 since the SIN response has both a transient and a sustained response and the transient response is similar in time to the LGN response. Other LGN or cortical neurons, which you are not recording from, are responsible for the sustained portion of the SIN response. One could say that the matching is incomplete but not that there is a mismatch as the time course of the LGN input is clearly expressed in the SIN response. Similarly with the RF sign overlap, the ON response of the SIN in Figure 2C is obviously caused by an ON LGN cell which you are not recording from, but there is a clear response to the OFF LGN cell. Under the authors definition of "mismatch" a SIN and an LGN neuron will never "match" since SINs have overlapping ON and OFF subregions.

3) Main issue concerning pre and postsynaptic components of the response:

The summary of the comments posted below concerns the use of the terms pre and postsynaptic. Essentially, the LFP amplitude is generated by postsynaptic processes, thus, describing it as presynaptic process is confusing. For example, changes in input resistance caused by changes in brain state would greatly change the amplitude of the LFP. An indirect measure of the presynaptic input would be provided by the sink caused by the invasion of synaptic terminals by incoming action potentials, which the authors showed in the past but did not provide here. Furthermore, response delay as a presumption of lack of LGN input is just as much pre as it is postsynaptic, in fact it refers to the same phenomenon as rule 2: size of LGN input, which in rule 3 is zero. Ultimately, both measures are indirect measures of connectivity and synaptic activation and the otherwise unambiguous terms pre and postsynaptic make the issue very confusing. All three reviewers view this terminology and the presentation of the results as oversimplifying and overselling conclusions since extracellular recordings of spikes and fields provide only indirect evidence of pre and postsynaptic measures, they do not provide any specific information on actual pre and postsynaptic processes. Thus, the authors should clarify the terminology throughout and do not utilize those terms. More importantly, the authors should pursue the analysis of the non-SIN neurons (Supplementary figure 2) to determine whether they also follow the three rules (see "e" below).

4) Issue of context:

The manuscript would be more appealing if it had a more general introduction, and a discussion that touches on the implications of their connectivity rules for sensory processing, cortical microcircuitry, and even the development of thalamocortical synapses onto SINs. Their data do provide some important clues about microcircuitry. There are, for example, SINs in their sample that do not receive monosynaptic thalamic input, yet the cells seem to reside in the termination zones of TC terminals. Something must determine whether TC axons connect or not to particular SINs, although the paper does not hint at anything mechanistic.

While the essential revisions are listed above, for the benefit of the authors we are pasting the detailed comments of the reviewers below in their entirety in case they find them useful in revising their resubmission:

a) Figure 3D shows a nice correlation between local spike-triggered-LFP amplitude and LGN-SIN efficacy. This is a beautiful result, but it raises a couple of questions of data analysis and semantics. If I understand the senior author's previous work on st-CSDs, the method identifies current sinks generated by both LGN intracortical axon spikes and sinks generated by LGN synapses upon cortical neurons (with the sinks perhaps generated mainly by vertically oriented pyramidal cells). If that's correct, then the only purely "presynaptic" component is the short-latency, brief, upwardly propagating axonal sink component. The longer latency, more localized component is the "postsynaptic" response. While the later sink obviously also depends on presynaptic components (synaptic functions require pre and post contributions, by definition), the sink is directly generated by postsynaptic mechanisms.

b) My questions: Does LGN-SIN efficacy correlate with any CSD measures? Figure 3D plots the peak postsynaptic LFP amplitude only, whereas the CSDs provide (in principle) additional spatial information (at least along the axis of the electrode array). CSDs also provide a true "presynaptic" measure, the early sinks, whereas the later sinks combine measures of pre- and postsynaptic functions. In the two example pairs shown in Figure 3, there is not only a big difference in the laminar locus of the LGN-triggered responses, but the axonal CSDs are very different in amplitude. This seems like a possible confound. Are the axonal sinks/spikes too small and unreliable to analyze across the cell sample? My first semantic quibble is about calling Rule 2 a "presynaptic factor", considering that the measure used to deduce it (the late L4 LFP) is largely a postsynaptic process.

c) To continue my pedantic semantic questions, why is Rule 3 a "postsynaptic factor", any more than Rule 2 is a "presynaptic factor"? Both designations seem to confuse the mechanistic issues. Rules 2 and 3 both involve different features of the same basic mechanism: either LGN axons form synapses on the SIN in question, or they don't. While the number and strength of those synapses are important factors, the methods used here cannot disambiguate those variables or infer anything exclusively pre- or postsynaptic about the connections. Rule 2 is about local connectivity (a SIN may make strong synapses with some LGN cells, but not with the cells being recorded in a particular experiment); rule 3 is about global connectivity (some SINs apparently don't get monosynaptic input from any LGN cells). This is an enormously important result, but I worry that the authors' semantic choices may obscure their conclusions for some readers. Labeling the rules "presynaptic" or "postsynaptic" is misleading, in my opinion. The problem is particularly acute in the Abstract and Introduction, where the rules are stated but only identified by their presynaptic/postsynaptic monikers and not further explained. The Discussion, by contrast, provides a very clear and compelling explanation of the authors' conclusions.

d) Although the results are very interesting they may be actually trivial, as the authors honestly mention in their Discussion. Indeed, rules 2 and 3 are probably simply spatial and temporal functional evidences of direct anatomical connection. Rule 2 is spatial and is a way to know whether the input arrives close to the recorded cell. Rule 3 is temporal and can be interpreted as expected delays between first-order and second-order cells (see Bullier et al., 1980 for instance). So if there is not the proper spatial conditions – the main input not in the right location – and/or not the proper temporal conditions – too long latency to be a first-order cell -, then it is entirely expected and logical that there is the connection is indirect. The results are really expected. Would rules 2 and 3 be linked to functional properties such as the RF structure or selectivity, as the authors also unsuccessfully investigated, the results would have had stronger impact because it would help understanding the relationship between the structure of the thalamo-cortical connectivity and the emergence of function. Currently, the benefit of these 3 rules is, as discussed, a methodological improvement for more unequivocal estimation of a direct connection: "This is a fairly easy measure to take in cross- correlation studies using extracellular microelectrodes, and we suggest that its general employment would reduce the variability seen in many cross-correlation studies of synaptic connectivity. " I agree it is an important methodological improvement for the field, but I feel not suited for *eLife*.

e) However, there may be one possibility, with the current dataset, to potentially increase the impact of this result. I suggest that the authors make a complete parallel analysis of the non-SIN neurons analysed in Supplementary figure 2. Adding the information on the retinotopic alignment (rule 1) and latency of the response (rule 3) will allow to state whether these neurons obey as well to the 3 rules or not. If they do, as is expected, then the 3 rules are really 3 independent functional measure of direct connection that can be applied to any neurons. If they don't, then it seems that the result is less trivial than expected, that will make it probably more interesting. The discussion of the potential circuitry and impact of such results on the thalamo-cortical connectivity would make it more suitable for *eLife*.

f) I don't really agree with the Abstract sentence "These rules concern (1) retinotopic alignment, (2) a property of the presynaptic axon, and (3) a property of the postsynaptic cortical interneuron". Rule 2 is not really a property of the presynaptic axon, which could be interpreted as conduction velocity for instance, but simply the laminar localization of the presynaptic terminals. The authors have not shown that rule 3 is a property of the cortical interneuron (integration time or Magno vs. Parvo cells for instance), since it can just be first vs. second-order cells. There are three major evidences in favor of the second hypothesis : 1) it is predictive of direct connection, 2) the authors haven't been able to relate the difference of connectivity to "sustained" or "transient" cells categories and 3) actually short and long latency SIN are undistinguishable functional properties. I think rule 2 would be better described as spatial connectivity profile and rule 3 has temporal properties that reflects 1st vs. 2nd order cells.

[Editors' note: further revisions were suggested prior to acceptance, as described below.]

Thank you for resubmitting your article "Three rules govern thalamocortical connectivity of fast-spike inhibitory interneurons in the visual cortex" for consideration by *eLife*. Your article has been reviewed by two peer reviewers, and the evaluation has been overseen by a Guest Reviewing Editor and John Huguenard as the Senior Editor. The following individual involved in review of your submission has agreed to reveal their identity: Barry Connors (Reviewer #2).

The reviewers have discussed the reviews with one another and the Reviewing Editor has drafted this decision to help you prepare a revised submission.

Summary:

The authors largely addressed all concerns raised in the first round of review. Overall, the impression remains positive both about the importance of the results and suitability for *eLife*. However, important concerns were raised that need to be addressed concerning language and presentation.

The main points are:

1) Better definition of the term "postsynaptic". See detailed explanation in reviewer 2 comment.

2) Clarification of SINs in comparison to layer 4 interneurons in other species. See comments by reviewer 1.

3) Enhancement of the main point, which is written in the response letter, that "virtually every first-order fast-spike interneuron in layer 4 receives input from nearly all of the LGN axons that synapse nearby (40/41 cases)". See comments by reviewer 3.

In addition to the written reviews, the three reviewers had a lengthy discussion regarding the revised manuscript. The reviewers praised the difficult and elegant technique used. Three key points of the discussion were:

1) The non-trivial nature of the high % of LGN inputs to SINs. It was noted that the main point that needs to come across to the reader is the sparseness of connections to RS cells, which are neighbors to the SINs and yet receive much less input (11% in this manuscript, but other examples in the literature are key to make the point). For those that have spent long hours (and years) chasing monosynaptic connections between thalamus and cortex, the numbers provided here are indeed surprising. However, without a clear notion of the difficulty and probability of finding connected pairs the three rules may appear rather trivial. The three reviewers agreed that in presenting the results, the authors should make an effort to convey this notion. When recording pairs, in which the cortical cell is an RS, a connection can be explained by the respective spatial and temporal properties of the two cells in the pair. But a lack of connection can almost never be explained. Why are thalamocortical axons so picky? The surprising finding here is that the authors are able to explain all cases in which there are NO connections based on the three rules. Which obviously means: 1) they can explain connectivity to all SINs and 2) SINs get a lot more LGN input than RS cells. These two points are novel and surprising and have convinced the reviewers of the suitability of the manuscript to *eLife*. However, it should be made clear also that a lot remains to be done to understand why RS cells are sparsely connected.

2) The perhaps unique nature of SINs. As indicated by reviewer 1, fast-spiking (FS) interneurons in cats and monkeys are simple cells with spatially separate RF subregions (the jury is still out on mice it seems). SINs in rabbits are, instead, complex cells. Perhaps they fulfill a unique functional role in rabbits, and for that receive many more connections. Or perhaps SINS and the FS cells typically identified in cat and monkey neurophysiological studies are distinctly different subtype of neurons. Can the authors provide more insight into the comparison of rabbit SINS and cat/monkey FS cells?

3) Emphasis and clarification must be placed in the existence of a group of SINs that do not receive any connection from LGN, which suggests the existence of two types of SINs with perhaps very different functional roles. The two main potential sources of excitatory input to SINS are LGN synapses and synapses from local spiny cells. Data suggest two groups of SINS: Most are driven by strong, convergent LGN input, and this makes them sensitive and visually nonspecific. Another small group receives no LGN input, yet they are still visually sensitive and nonspecific, suggesting they are largely driven by strong, convergent input from local spiny cells, which is entirely consistent with a variety of anatomical and paired-cell in vitro recording studies that the authors do not cite, local FS interneurons in general have high probabilities of connectivity (~0.5) with spiny neurons in their local neighborhood.

This point was also viewed by the reviewers as a key and novel result that needs emphasis and clarification.

Reviewer #1:

An important notion that needs to be made very clear in the manuscript is that there are no interneurons in layer 4 of cat or monkey V1 that look like SINs. The vast majority of neurons in thalamorecipient layer 4 are simple cells, i.e. they have segregated ON and OFF subregions (apart from center surround neurons in monkey which are not relevant to this Discussion). In contrast, SINs are complex cells with overlapping subregions of opposing contrast. That makes SIN a unique category of cells and perhaps that is the reason why they have such large percentage of input compared to cat L4 neurons. Based on that, comparisons of LGN % connectivity from other species are only mildly informative and the best comparison is with neurons in rabbit cortex itself. (which the authors don't fully provide).

Comparisons with barrel cortex are tricky because the functional properties that may determine convergence are not well understood, although sparse connectivity of VB to barrel neurons does enhance the uniqueness of SINs notoriously promiscuous connectivity.

So the authors show that, if there is RF overlap (rule1) and thalamic input to the vicinity of the SIN (st-LFP, rule2), the SIN will receive a connection, unless the SIN does not receive thalamic input at all (e-stim,rule3). Thus, SINs receive a lot more input than RS cells, which, under the same circumstances, only receive a small percentage of the available thalamic input. Sure, this is interesting and important, but by itself it does not create a universal new knowledge about L4 gabaergic interneurons because of the uniqueness of SINs. Comparing their promiscuous input to cat or monkey L4 interneurons is incorrect as in those two species FS neurons in layer 4 are simple cells with segregated subregions and are selective to orientation, direction and spatial frequency (SINs are very broadly tuned as shown by the authors in Zhuang et al). The broader tuning properties of layer 4 FS cells in cat and monkey is no match for the SINs.

Reviewer #2:

The authors have done an excellent job of responding to the reviewers' comments. Their revised manuscript provides clearer explanations of the "three rules", and new additions to the Introduction and Discussion broaden the context and appeal of the results.

I have one small suggestion. While the authors were very responsive to the reviewers' concerns about the meanings of "presynaptic" and "postsynaptic" in the original manuscript, the revision still has a phrase that will (in my opinion) confuse many readers. It first appears in the Abstract as a summary of rule #2: "strength of the postsynaptic response elicited near the interneuron by the LGN neuron", and again in a slightly different form in the Introduction.

Confusion will arise because is unclear In the Abstract and Introduction what "postsynaptic response" is referred to. Most readers' default assumption will be that it refers to the response of the postsynaptic neuron under study (i.e. a SIN), and the readers' own response might be to think, "well, of course you need a strong postsynaptic response from a SIN to consider it connected to the LGN cell". That's obvious and uninteresting.

But actually, the authors are referring to the "spike-triggered postsynaptic LFP response generated by the LGN neuron" (spelled out clearly much later in the paper). In other words, the authors' "postsynaptic measure, i.e. an st-LFP (or CSD), is derived not from the SIN under study but from other neighboring unidentified neurons; another reasonable interpretation of this measure is that the st-LFP/CSD is an estimate of the strength and location of the presynaptic terminals of the single LGN cell under study.

Bottom line: I think the authors need to find a better way to state their rule #2 early in the paper if they want their readers to appreciate the point.

Reviewer #3:

In their revised version the authors addressed correctly and convincingly most of the concerns of the first review and the papers gains globally clarity. There remains a substantive concern regarding the analysis of non-SIN neurons

The authors have explained lacking experimental data to fully address the question and this is granted. However, can the authors be less conservative ?

It is understood that the authors do not have enough data to check for the 1st and 3rd rule. However, the result for rule 2 is quite different: only 11% of them are connected compared to 73% for the SIN, comparing Figure 3D with 5A is quite convincing that nonSIN and SIN behave differently.

This would be important to help the author convince the reader that the behavior of non-SIN are different and hereby be more convincing that their result is not trivial.

What they answered to this concern, and their discussion is not fully convincing in the current state. In their answer, the authors stated that : “these (3 rules) "necessary conditions" are the ONLY requirements (other than retinotopic alignment) that must be fulfilled to generate a functional synaptic connection. That these three requirements are both necessary and sufficient is new, and that is surprising.”

It is new indeed, but at first glance do not seem surprising (see below).

The authors further argue that:

“All previous measures of thalamocortical connectivity indicated that only a subset of cortical neurons located within the axonal field of a thalamic afferent receives connection from that afferent. This is true for regular spiking neurons in the cat visual cortex (15%-33%, Sedigh-Sarvestani et al., 2017; Alonso et al., 2001), regular spiking neurons in rat barrel cortex (37%, Bruno and Simons, 2002) and fast spiking neurons in rat barrel cortex (63%, Bruno and Simons, 2002). This specific wiring is also true for other stages of sensory processing (e.g. 4/20, 20% for retinogeniculate connections in cats, Hamos et al., 1987). Given the strong evidence from the past that geniculate afferents make connection with only a subset of cortical neurons, our results are clearly surprising”

This argument concerns rule 2 only and demonstrate that, in itself, it is not sufficient. So further constrains must exist, as a consequence it does not make their result surprising, but rather expected at first glance. And alignment of RF and first-order cell are not the most surprising parameters to look at for further constrains.

Their last sentence “This finding is clearly unexpected and has major consequences for understanding how thalamocortical inhibitory networks develop and are functionally organized.” seems like an overstatement.

Note from editor: this comment was made prior to the discussion among the reviewers. The reviewers have agreed on the suitability for *eLife* upon discussion. To make the current manuscript more convincing and suitable for *eLife*, I believe the authors have more work to do. First, maybe by exploiting more the large difference between Figure 3D and 5A. Second, by improving their discussion on that point that is not clear and not as convincing as their reply to our first review (“That these three requirements are both necessary and sufficient is new, and that is surprising”):

It is not clear what is the logical flow that leads to some sentences, for instance “Under this scheme, excitatory cortical neurons become selective to sensory stimuli through two complementary mechanisms: a selected pool of thalamic inputs that build the basic receptive field structure and a nonselective feed-forward inhibitory network that sharpens receptive field selectivity.”. They speak about “different type of fast spiking cells” which is misleading (what “types” are they referring to ?). “Each thalamic afferent makes connection with nearly all of the first-order fast-spike inhibitory neurons that are within reach” is a circular argument, if they make connection they are first-order cells, this argument could be better written.

---

## [Author Response]

Revisions:1) The three reviewers agreed that the article is clearly written, well-documented and illustrated and very pleasant to read, the methodology is powerful and flawless, the results are convincing. The study represents another important step forward in the understanding of thalamocortical transmission of sensory signals to neocortical layer 4. However, the three reviewers agree on the confusing nature of the terminology pre and postsynaptic and the need for additional analysis of non-SIN neurons.

We thank the reviewers for their careful and generally positive review of our work. We believe that we have complied with all reviewers’ requests, save one, the request for a “complete parallel analysis of the non-SIN neurons”. Although we have done some further analyses of these data (below, new Figure. 5B) We feel that we cannot do the “complete parallel analysis” requested without extensive further experimentation This is because we have limited receptive-field measures on these cells, little data on the latency of their responses to thalamic electrical stimulation and visual stimulation, and a very limited number that were connected (n = 3, Figure. 5). Please see our further discussion of this below.

2) Issue regarding terminology:One reviewer is concerned that the term mismatch applied to the sign of the RF and the time course of the response seems misleading. For example, I don't see a mismatch in time course in Figure 2 since the SIN response has both a transient and a sustained response and the transient response is similar in time to the LGN response. Other LGN or cortical neurons, which you are not recording from, are responsible for the sustained portion of the SIN response. One could say that the matching is incomplete but not that there is a mismatch as the time course of the LGN input is clearly expressed in the SIN response. Similarly with the RF sign overlap, the ON response of the SIN in Figure 2C is obviously caused by an ON LGN cell which you are not recording from, but there is a clear response to the OFF LGN cell. Under the authors definition of "mismatch" a SIN and an LGN neuron will never "match" since SINs have overlapping ON and OFF subregions.

We agree with the reviewer’s argument that the terms “match” and “mismatch” are too strong for describing the ON/OFF agreement between LGN neurons and SINs, because LGN cells are either ON or OFF centered, but SINs have both ON and OFF responses. However, SINs may be ON-dominated or OFF-dominated, and we do maintain that it is appropriate to say that the response properties of an ON center LGN response and an ON-Dominated SIN are “more similar” than are the response properties of an ON-center LGN cell and an OFF-dominated SIN. Therefore, we have eliminated the match/mismatch terms in favor of the terms similarity/dissimilarity (a less dichotomous distinction).

We have done the same for sustained vs. transient responding. Please note, however, that there is a 40+ year history in the literature classifying visual neurons (Retina, LGN and cortex) as sustained or transient based on the presence or absence of a sustained response component (e.g. Cleland, Levick and Sanderson, J. Physiol,, 1973; Cleland et al., 1976, Bezdudnaya et al., 2006; Cano et al., 2006; Stoelzel et al., 2008; Zhuang et al., 2013). In these schemes, “transient” cells have only a transient response component, while “sustained” cells may have both.

3) Main issue concerning pre and postsynaptic components of the response:The summary of the comments posted below concerns the use of the terms pre and postsynaptic. Essentially, the LFP amplitude is generated by postsynaptic processes, thus, describing it as presynaptic process is confusing. For example, changes in input resistance caused by changes in brain state would greatly change the amplitude of the LFP. An indirect measure of the presynaptic input would be provided by the sink caused by the invasion of synaptic terminals by incoming action potentials, which the authors showed in the past but did not provide here. Furthermore, response delay as a presumption of lack of LGN input is just as much pre as it is postsynaptic, in fact it refers to the same phenomenon as rule 2: size of LGN input, which in rule 3 is zero. Ultimately, both measures are indirect measures of connectivity and synaptic activation and the otherwise unambiguous terms pre and postsynaptic make the issue very confusing. All three reviewers view this terminology and the presentation of the results as oversimplifying and overselling conclusions since extracellular recordings of spikes and fields provide only indirect evidence of pre and postsynaptic measures, they do not provide any specific information on actual pre and postsynaptic processes. Thus, the authors should clarify the terminology throughout and do not utilize those terms.

We agree completely that this is unduly confusing and have made the appropriate changes throughout the manuscript.

More importantly, the authors should pursue the analysis of the non-SIN neurons (Supplementary figure 2) to determine whether they also follow the three rules (see "e" below).

We would like very much to fully comply with this this important suggestion. However, we do not have enough data on the non-SINs. Importantly, only three non-SINs (3/28, ~11%) showed connectivity (Supplementary figure 2), despite the fact that they were very near (< 150 μm vertically) to SINs that were connected to the same LGN axon, and the spike-triggered postsynaptic LFPs elicited by the same LGN neuron were nearly as strong near the non-SIN as they were near the SINs (see new Figure. 5B and associated text). Moreover, we did not record the response latencies to electrical or visual stimulation for many of the non-SINs, or study their receptive fields in sufficient detail. Therefore, we would not, without extensive further data collection, be able to determine whether our “rules” apply to these cells as they do for the SINs. We are hopeful that the reviewers would agree with our conservative view: That there is, at present, simply not enough relevant data on the non-SINs to make convincing arguments beyond the very limited statements that we made (i.e. that regular-spiking non-SINs were less frequently connected than were SINs that were found in the same terminal LGN terminal field). We emphasize, however, that the main point of our study is not simply that layer 4 fast-spike interneurons are more connected to the thalamus than are the spiny regular spiking neurons. We believe that our most important finding is that virtually every first-order fast-spike interneuron in layer 4 receives input from nearly all of the LGN axons that synapse nearby (40/41 cases).

4) Issue of context:The manuscript would be more appealing if it had a more general introduction, and a discussion that touches on the implications of their connectivity rules for sensory processing, cortical microcircuitry, and even the development of thalamocortical synapses onto SINs. Their data do provide some important clues about microcircuitry. There are, for example, SINs in their sample that do not receive monosynaptic thalamic input, yet the cells seem to reside in the termination zones of TC terminals. Something must determine whether TC axons connect or not to particular SINs, although the paper does not hint at anything mechanistic.

We have modified the Introduction and Discussion in line with this suggestion.

While the essential revisions are listed above, for the benefit of the authors we are pasting the detailed comments of the reviewers below in their entirety in case they find them useful in revising their resubmission:a) Figure 3D shows a nice correlation between local spike-triggered-LFP amplitude and LGN-SIN efficacy. This is a beautiful result, but it raises a couple of questions of data analysis and semantics. If I understand the senior author's previous work on st-CSDs, the method identifies current sinks generated by both LGN intracortical axon spikes and sinks generated by LGN synapses upon cortical neurons (with the sinks perhaps generated mainly by vertically oriented pyramidal cells). If that's correct, then the only purely "presynaptic" component is the short-latency, brief, upwardly propagating axonal sink component. The longer latency, more localized component is the "postsynaptic" response. While the later sink obviously also depends on presynaptic components (synaptic functions require pre and post contributions, by definition), the sink is directly generated by postsynaptic mechanisms.

We agree that the observed later sink is generated purely by postsynaptic mechanisms. However, presynaptic terminals are necessary for this postsynaptic sink and our thinking was that this fast postsynaptic sink can be used to infer the density of presynaptic terminals in the area. Nevertheless, we agree that this is unnecessarily confusing, so we have changed this terminology throughout the manuscript, as described above (and suggested by the reviewers)

b) My questions: Does LGN-SIN efficacy correlate with any CSD measures? Figure 3D plots the peak postsynaptic LFP amplitude only, whereas the CSDs provide (in principle) additional spatial information (at least along the axis of the electrode array). CSDs also provide a true "presynaptic" measure, the early sinks, whereas the later sinks combine measures of pre- and postsynaptic functions. In the two example pairs shown in Figure 3, there is not only a big difference in the laminar locus of the LGN-triggered responses, but the axonal CSDs are very different in amplitude. This seems like a possible confound. Are the axonal sinks/spikes too small and unreliable to analyze across the cell sample? My first semantic quibble is about calling Rule 2 a "presynaptic factor", considering that the measure used to deduce it (the late L4 LFP) is largely a postsynaptic process.

As described in Figure 1, our cortical recordings were made either on single electrodes or on a laminar probe. Since the axonal and postsynaptic LFPs can be measured on single electrodes (Swadlow and Gusev, 2000) and on the laminar probes, they provided a more complete data set than did the CSD measures. Notably, the axonal responses (at the same depth as the SIN under study) is poorly correlated with synaptic efficacy (R2 = 0.0011). This is because the axonal and the postsynaptic responses often reach a peak at different depths. Sometimes the axonal sinks can be seen deep in the cortex at earliest latency (where there is no clear postsynaptic sink, as in our Figure 3B1). Indeed, sometimes there is a strong axonal response seen ascending through the cortex, but no postsynaptic response (e.g. Jin et al., 2011, Figure 1C case on right). Presumably, this case represents recordings from an ascending axon near our probe that turns laterally in the cortex and then arborizes and forms synapses (away from the recording probe).

We agree with the “first semantic quibble” and have changed the description of our “rules” throughout the manuscript.

c) To continue my pedantic semantic questions, why is Rule 3 a "postsynaptic factor", any more than Rule 2 is a "presynaptic factor"? Both designations seem to confuse the mechanistic issues. Rules 2 and 3 both involve different features of the same basic mechanism: either LGN axons form synapses on the SIN in question, or they don't. While the number and strength of those synapses are important factors, the methods used here cannot disambiguate those variables or infer anything exclusively pre- or postsynaptic about the connections. Rule 2 is about local connectivity (a SIN may make strong synapses with some LGN cells, but not with the cells being recorded in a particular experiment); rule 3 is about global connectivity (some SINs apparently don't get monosynaptic input from any LGN cells). This is an enormously important result, but I worry that the authors' semantic choices may obscure their conclusions for some readers. Labeling the rules "presynaptic" or "postsynaptic" is misleading, in my opinion. The problem is particularly acute in the Abstract and Introduction, where the rules are stated but only identified by their presynaptic/postsynaptic monikers and not further explained. The Discussion, by contrast, provides a very clear and compelling explanation of the authors' conclusions.

We agree (as described above) and have changed the text and figures appropriately.

d) Although the results are very interesting they may be actually trivial, as the authors honestly mention in their discussion. Indeed, rules 2 and 3 are probably simply spatial and temporal functional evidences of direct anatomical connection. Rule 2 is spatial and is a way to know whether the input arrives close to the recorded cell. Rule 3 is temporal and can be interpreted as expected delays between first-order and second-order cells (see Bullier et al., 1980 for instance). So if there is not the proper spatial conditions – the main input not in the right location – and/or not the proper temporal conditions – too long latency to be a first-order cell -, then it is entirely expected and logical that there is the connection is indirect. The results are really expected. Would rules 2 and 3 be linked to functional properties such as the RF structure or selectivity, as the authors also unsuccessfully investigated, the results would have had stronger impact because it would help understanding the relationship between the structure of the thalamo-cortical connectivity and the emergence of function. Currently, the benefit of these 3 rules is, as discussed, a methodological improvement for more unequivocal estimation of a direct connection: "This is a fairly easy measure to take in cross- correlation studies using extracellular microelectrodes, and we suggest that its general employment would reduce the variability seen in many cross-correlation studies of synaptic connectivity. " I agree it is an important methodological improvement for the field, but I feel not suited for eLife.

We respectfully disagree. Of course, for connectivity to occur, presynaptic terminals must be nearby (rule 2), and the postsynaptic neuron must be receptive to some LGN input (rule 3). Our ability to measure these variables while recording in vivo is new, and we agree it is a nice methodological contribution. However, what is new and surprising (as a conceptual contribution) is that these “necessary conditions” are the ONLY requirements (other than retinotopic alignment) that must be fulfilled to generate a functional synaptic connection. That these three requirements are both necessary and sufficient is new, and that is surprising.

All previous measures of thalamocortical connectivity indicated that only a subset of cortical neurons located within the axonal field of a thalamic afferent receives connection from that afferent. This is true for regular spiking neurons in the cat visual cortex (15%-33%, Sedigh-Sarvestani et al., 2017; Alonso et al., 2001), regular spiking neurons in rat barrel cortex (37%, Bruno and Simons, 2002) and fast spiking neurons in rat barrel cortex (63%, Bruno and Simons, 2002). This specific wiring is also true for other stages of sensory processing (e.g. 4/20, 20% for retinogeniculate connections in cats, Hamos et al., 1987). Given the strong evidence from the past that geniculate afferents make connection with only a subset of cortical neurons, our results are clearly surprising. They indicate that EVERY geniculate axon makes connection with ALL first-order fast spiking cortical neurons that are within reach of the axonal arbor (first-order defined as inhibitory neurons that receive thalamic input). This finding is clearly unexpected and has major consequences for understanding how thalamocortical inhibitory networks develop and are functionally organized.

e) However, there may be one possibility, with the current dataset, to potentially increase the impact of this result. I suggest that the authors make a complete parallel analysis of the non-SIN neurons analysed in Supplementary figure 2. Adding the information on the retinotopic alignment (rule 1) and latency of the response (rule 3) will allow to state whether these neurons obey as well to the 3 rules or not. If they do, as is expected, then the 3 rules are really 3 independent functional measure of direct connection that can be applied to any neurons. If they don't, then it seems that the result is less trivial than expected, that will make it probably more interesting. The discussion of the potential circuitry and impact of such results on the thalamo-cortical connectivity would make it more suitable for eLife.

As noted above, we would like to comply more fully with this request, but we don’t have enough data on non-SINs to make meaningful conclusions other than the single conclusion that we reached: that they were less frequently connected than were the SINs, despite being in the synaptic field of the same LGN neuron that was driving a neighboring SIN.

f) I don't really agree with the Abstract sentence "These rules concern (1) retinotopic alignment, (2) a property of the presynaptic axon, and (3) a property of the postsynaptic cortical interneuron". Rule 2 is not really a property of the presynaptic axon, which could be interpreted as conduction velocity for instance, but simply the laminar localization of the presynaptic terminals. The authors have not shown that rule 3 is a property of the cortical interneuron (integration time or Magno vs. Parvo cells for instance), since it can just be first vs. second-order cells. There are three major evidences in favor of the second hypothesis : 1) it is predictive of direct connection, 2) the authors haven't been able to relate the difference of connectivity to "sustained" or "transient" cells categories and 3) actually short and long latency SIN are undistinguishable functional properties. I think rule 2 would be better described as spatial connectivity profile and rule 3 has temporal properties that reflects 1st vs. 2nd order cells.

We agree (see above) and have changed the wording of the “rules”.

[Editors' note: further revisions were suggested prior to acceptance, as described below.]

The authors largely addressed all concerns raised in the first round of review. Overall, the impression remains positive both about the importance of the results and suitability for eLife. However, important concerns were raised that need to be addressed concerning language and presentation.The main points are:1) Better definition of the term "postsynaptic". See detailed explanation in reviewer 2 comment.

We have changed our definition in the Abstract and throughout the manuscript, in line with the reviewers’ suggestion.

2) Clarification of SINs in comparison to layer 4 interneurons in other species. See comments by reviewer 1.

We have added a section on this in the Discussion entitled “Comparing the receptive fields of rabbit L4 SINs with those of other species”.

3) Enhancement of the main point, which is written in the response letter, that "virtually every first-order fast-spike interneuron in layer 4 receives input from nearly all of the LGN axons that synapse nearby (40/41 cases)". See comments by reviewer 3.

We have highlighted this idea in the Abstract (“We conclude that virtually every first-order fast-spike interneuron in layer 4 receives input from nearly all of the LGN axons that synapse nearby.”) and elsewhere in the text. In addition, we added a new section in Discussion where this is discussed further relative to the sparseness of connections to regular spiking cells: “Comparisons with L4 regular spiking (presumptive spiny) neurons”.

In addition to the written reviews, the three reviewers had a lengthy discussion regarding the revised manuscript. The reviewers praised the difficult and elegant technique used. Three key points of the discussion were:1) The non-trivial nature of the high % of LGN inputs to SINs. It was noted that the main point that needs to come across to the reader is the sparseness of connections to RS cells, which are neighbors to the SINs and yet receive much less input (11% in this manuscript, but other examples in the literature are key to make the point). For those that have spent long hours (and years) chasing monosynaptic connections between thalamus and cortex, the numbers provided here are indeed surprising. However, without a clear notion of the difficulty and probability of finding connected pairs the three rules may appear rather trivial. The three reviewers agreed that in presenting the results, the authors should make an effort to convey this notion. When recording pairs, in which the cortical cell is an RS, a connection can be explained by the respective spatial and temporal properties of the two cells in the pair. But a lack of connection can almost never be explained. Why are thalamocortical axons so picky? The surprising finding here is that the authors are able to explain all cases in which there are NO connections based on the three rules. Which obviously means: 1) they can explain connectivity to all SINs and 2) SINs get a lot more LGN input than RS cells. These two points are novel and surprising and have convinced the reviewers of the suitability of the manuscript to eLife. However, it should be made clear also that a lot remains to be done to understand why RS cells are sparsely connected.

To better emphasize this, we have added a section to the Discussion “Comparisons with L4 regular spiking (presumptive spiny) neurons”.

2) The perhaps unique nature of SINs. As indicated by reviewer 1, fast-spiking (FS) interneurons in cats and monkeys are simple cells with spatially separate RF subregions (the jury is still out on mice it seems). SINs in rabbits are, instead, complex cells. Perhaps they fulfill a unique functional role in rabbits, and for that receive many more connections. Or perhaps SINS and the FS cells typically identified in cat and monkey neurophysiological studies are distinctly different subtype of neurons. Can the authors provide more insight into the comparison of rabbit SINS and cat/monkey FS cells?

We respectfully disagree with the reviewer on this issue and have added a section to the Discussion concerning such comparisons: “Comparing the receptive fields of rabbit L4 SINs with those of other species”.

3) Emphasis and clarification must be placed in the existence of a group of SINs that do not receive any connection from LGN, which suggests the existence of two types of SINs with perhaps very different functional roles. The two main potential sources of excitatory input to SINS are LGN synapses and synapses from local spiny cells. Data suggest two groups of SINS: Most are driven by strong, convergent LGN input, and this makes them sensitive and visually nonspecific. Another small group receives no LGN input, yet they are still visually sensitive and nonspecific, suggesting they are largely driven by strong, convergent input from local spiny cells which is entirely consistent with a variety of anatomical and paired-cell in vitro recording studies that the authors do not cite, local FS interneurons in general have high probabilities of connectivity (~0.5) with spiny neurons in their local neighborhood.This point was also viewed by the reviewers as a key and novel result that needs emphasis and clarification.

We have added a new section to the Discussion: “The synthesis of receptive fields in L4 SINs”) which contains a paragraph dealing these “long-latency” SINs.

Reviewer #1:An important notion that needs to be made very clear in the manuscript is that there are no interneurons in layer 4 of cat or monkey V1 that look like SINs. The vast majority of neurons in thalamorecipient layer 4 are simple cells, i.e. they have segregated ON and OFF subregions (apart from center surround neurons in monkey which are not relevant to this discussion). In contrast, SINs are complex cells with overlapping subregions of opposing contrast. That makes SIN a unique category of cells and perhaps that is the reason why they have such large percentage of input compared to cat L4 neurons. Based on that, comparisons of LGN % connectivity from other species are only mildly informative and the best comparison is with neurons in rabbit cortex itself. (which the authors don't fully provide).Comparisons with barrel cortex are tricky because the functional properties that may determine convergence are not well understood, although sparse connectivity of VB to barrel neurons does enhance the uniqueness of SINs notoriously promiscuous connectivity.So the authors show that, if there is RF overlap (rule1) and thalamic input to the vicinity of the SIN (st-LFP, rule2), the SIN will receive a connection, unless the SIN does not receive thalamic input at all (e-stim,rule3). Thus, SINs receive a lot more input than RS cells, which, under the same circumstances, only receive a small percentage of the available thalamic input. Sure, this is interesting and important, but by itself it does not create a universal new knowledge about L4 gabaergic interneurons because of the uniqueness of SINs. Comparing their promiscuous input to cat or monkey L4 interneurons is incorrect as in those two species FS neurons in layer 4 are simple cells with segregated subregions and are selective to orientation, direction and spatial frequency (SINs are very broadly tuned as shown by the authors in Zhuang et al). The broader tuning properties of layer 4 FS cells in cat and monkey is no match for the SINs.

We disagree strongly (but very respectfully!) with that statement, and explain our thinking in the new Discussion section entitled “ Comparing the receptive field of rabbit L4 SINs with those of other species ”.

Reviewer #2:The authors have done an excellent job of responding to the reviewers' comments. Their revised manuscript provides clearer explanations of the "three rules", and new additions to the Introduction and Discussion broaden the context and appeal of the results.I have one small suggestion. While the authors were very responsive to the reviewers' concerns about the meanings of "presynaptic" and "postsynaptic" in the original manuscript, the revision still has a phrase that will (in my opinion) confuse many readers. It first appears in the Abstract as a summary of rule #2: "strength of the postsynaptic response elicited near the interneuron by the LGN neuron", and again in a slightly different form in the Introduction.Confusion will arise because is unclear In the Abstract and Introduction what "postsynaptic response" is referred to. Most readers' default assumption will be that it refers to the response of the postsynaptic neuron under study (i.e. a SIN), and the readers' own response might be to think, "well, of course you need a strong postsynaptic response from a SIN to consider it connected to the LGN cell". That's obvious and uninteresting.But actually, the authors are referring to the "spike-triggered postsynaptic LFP response generated by the LGN neuron" (spelled out clearly much later in the paper). In other words, the authors' "postsynaptic measure, i.e. an st-LFP (or CSD), is derived not from the SIN under study but from other neighboring unidentified neurons; another reasonable interpretation of this measure is that the st-LFP/CSD is an estimate of the strength and location of the presynaptic terminals of the single LGN cell under study.Bottom line: I think the authors need to find a better way to state their rule #2 early in the paper if they want their readers to appreciate the point.

Thank you, we agree and have restated this “rule” in the Abstract, and everywhere it comes up in the paper.

Reviewer #3:In their revised version the authors addressed correctly and convincingly most of the concerns of the first review and the papers gains globally clarity. There remains a substantive concern regarding the analysis of non-SIN neuronsThe authors have explained lacking experimental data to fully address the question and this is granted. However, can the authors be less conservative ?It is understood that the authors do not have enough data to check for the 1st and 3rd rule. However, the result for rule 2 is quite different: only 11% of them are connected compared to 73% for the SIN, comparing Figure 3D with 5A is quite convincing that nonSIN and SIN behave differently.This would be important to help the author convince the reader that the behavior of non-SIN are different and hereby be more convincing that their result is not trivial.

We have tried to be “less conservative” in this revision and have included a new section in the Discussion that concerns the “The sparseness of TC connections to L4 regular spiking (presumptive spiny) neurons”. In that section, as the reviewer suggests, we direct the reader to “compare Figures 5A and 3D”. We also now note the RS results in the Abstract.

What they answered to this concern, and their discussion is not fully convincing in the current state. In their answer, the authors stated that : “these (3 rules) "necessary conditions" are the ONLY requirements (other than retinotopic alignment) that must be fulfilled to generate a functional synaptic connection. That these three requirements are both necessary and sufficient is new, and that is surprising.”It is new indeed, but at first glance do not seem surprising (see below).The authors further argue that:“All previous measures of thalamocortical connectivity indicated that only a subset of cortical neurons located within the axonal field of a thalamic afferent receives connection from that afferent. This is true for regular spiking neurons in the cat visual cortex (15%-33%, Sedigh-Sarvestani et al., 2017; Alonso et al., 2001), regular spiking neurons in rat barrel cortex (37%, Bruno and Simons, 2002) and fast spiking neurons in rat barrel cortex (63%, Bruno and Simons, 2002). This specific wiring is also true for other stages of sensory processing (e.g. 4/20, 20% for retinogeniculate connections in cats, Hamos et al., 1987). Given the strong evidence from the past that geniculate afferents make connection with only a subset of cortical neurons, our results are clearly surprising”

Regarding our “surprise” to find that the three rules were necessary and sufficient to predict TC connectivity, we now state that: “The “three rules” of synaptic connectivity that we describe are both necessary and sufficient for predicting connectivity between LGN neurons and L4 SINs (40/41 cases, Figure 4C). Together, they predict connection probability with a surprising high level of precision that is not yet possible for any other thalamocortical circuit in any other species.”

This argument concerns rule 2 only and demonstrate that, in itself, it is not sufficient. So further constrains must exist, as a consequence it does not make their result surprising, but rather expected at first glance. And alignment of RF and first-order cell are not the most surprising parameters to look at for further constrains.Their last sentence “This finding is clearly unexpected and has major consequences for understanding how thalamocortical inhibitory networks develop and are functionally organized.” seems like an overstatement.

We have removed that statement.